# Gaps in Network Infrastructure limit our understanding of biogenic methane emissions for the United States

Sparkle L. Malone[1], Youmi Oh[2], Kyle A. Arndt[3], George Burba[4,5], Roisin Commane[6], Alexandra R. Contosta[3], Jordan P. Goodrich[7], Henry W. Loescher[8,9], Gregory Starr[10], Ruth K. Varner[3,11]

[1] Institute of the Environment & Department of Biological Sciences, Florida International University, 11200 S.W. 8th Street, Miami, FL 33199, USA
[2] Cooperative Institute for Research in Environmental Sciences, University of Colorado, Boulder, CO 80309, United States
[3] Earth Systems Research Center, Institute for the Study of Earth, Oceans, and Space, University of New Hampshire, 8 College Rd, Durham, NH, 03824, USA
[4] LI-COR Biosciences, 4421 Superior St., Lincoln, NE, 68504, USA
[5] The Robert B. Daugherty Water for Food Global Institute and School of Natural Resources, University of Nebraska, Lincoln, NE, 68583
[6] Department of Earth & Environmental Sciences, Lamont-Doherty Earth Observatory, Columbia University, Palisades, NY 10964, USA
[7] School of Science, University of Waikato, Gate 1 Knighton Rd, Hillcrest 3240, Hamilton, New Zealand
[8] Battelle, National Ecological Observatory Network (NEON), Boulder, CO USA 80301
[9] Institute of Alpine and Arctic Research, University of Colorado, Boulder, CO USA 80301
[10] Department of Biological Sciences, University of Alabama, Tuscaloosa, AL. 35487, USA
[11] Department of Earth Sciences, University of New Hampshire, 56 College Rd, Durham, NH, 03824, USA

*Correspondence to*: Sparkle L. Malone (smalone@fiu.edu)

**Abstract.** Understanding the sources and sinks of methane ($CH_4$) is critical to both predicting and mitigating future climate change. There are large uncertainties in the global budget of atmospheric $CH_4$, but natural emissions are estimated to be of a similar magnitude to anthropogenic emissions. To understand $CH_4$ flux from biogenic sources in the United States (US) of America, a multi-scale $CH_4$ observation network focused on $CH_4$ flux rates, processes, and scaling methods is required. This can be achieved with a network of ground-based observations that are distributed based on climatic regions and landcover. To determine the gaps in physical infrastructure for developing this network, we need to understand the landscape representativeness of the current infrastructure. We focus here on eddy covariance (EC) flux towers because they are essential for a bottom-up framework that bridges the gap between point-based chamber measurements and airborne or satellite platforms that inform policy decisions and global climate agreements. Using dissimilarity, multidimensional scaling, and cluster analysis, the US was divided into 10 clusters distributed across temperature and precipitation gradients. We evaluated dissimilarity within each cluster for research sites with active $CH_4$ EC towers to identify gaps in existing infrastructure that limit our ability to constrain the contribution of US biogenic $CH_4$ emissions to the global budget. Through our analysis using climate, land cover, and location variables, we identified priority areas for research infrastructure to provide a more complete understanding of the $CH_4$ flux potential of ecosystem types across the US. Clusters corresponding to Alaska and the Rocky Mountains, that are inherently difficult to capture, are the most poorly represented and all clusters require a greater representation of vegetation types.

## 1 Introduction

The 21st century is characterized by ongoing changes in Earth's climate system that result from increasing concentrations of radiatively important trace gases in the atmosphere. Unlike the relatively steady increases of atmospheric carbon dioxide ($CO_2$) and nitrous oxide ($N_2O$), atmospheric methane ($CH_4$) concentrations show dynamic trends with a rapid increase of ~10 ppb yr$^{-1}$ since 2014 (Nisbet et al., 2019). The annual increase of atmospheric $CH_4$ in 2020 was the largest on record at ~15 ppb yr$^{-1}$ (Dlugokencky, 2021), despite the global pandemic reducing energy demand (Le Quéré et al., 2021). Increasing atmospheric $CH_4$ concentrations (Nisbet et al., 2019) is of concern because $CH_4$ is 34 times more effective at trapping heat in the atmosphere compared to an equivalent mass of $CO_2$ over a 100-year timeframe, and accounts for ~42% of warming since the pre-industrial period (IPCC, 2021). These rapid increases in atmospheric $CH_4$ challenge us to reach the goals of the Paris Agreement (Nisbet et al., 2019) but also provide an opportunity given the relatively short atmospheric residence time (~9 years) of $CH_4$. Understanding the sources and sinks of $CH_4$ is therefore critical in predicting and mitigating future climate change.

Quantifying the national $CH_4$ budget is important for assessing realistic pathways to mitigate climate change, yet uncertainties in the magnitude, size, and location of sources and sinks limit budget development (Saunois et al., 2020; Bruhwiler et al., 2021). Methane is emitted from a variety of often co-located biogenic, thermogenic, and pyrogenic sources (IPCC, 2013; Nisbet et al., 2019). Biogenic emissions are thought to be of a similar magnitude to total anthropogenic emissions, yet biogenic $CH_4$ emissions remain the most uncertain source of the global $CH_4$ budget (Saunois et al., 2020). Surface-atmosphere exchange from biogenic sources and sinks, the biological and environmental processes driving these fluxes (e.g., ebullition, aerenchyma pumping), and how $CH_4$ sources and sinks change over space and time, including interannual variability (Michalak et al., 2009; Kirschke et al., 2013; Knox et al., 2019; Nisbet et al., 2019), are not well constrained. Finally, both the vast areas with relatively small uptake and emission rates (e.g., deserts, grasslands, forests) and the lake-ocean water continuum that transports $CH_4$ (e.g., fens, streams, and rivers) have been largely understudied but could contribute significantly to regional and global budgets (Hutchins et al., 2019; Rosentreter et al., 2021; Zhou et al., 2021). These unknowns hinder our ability to predict future climate change due to the complex feedbacks between biological processes (e.g., microbial production and consumption) (Sherwood et al., 2017; Zhang et al., 2017; Oh et al., 2020), climate change (Zhang et al., 2017), and land cover change (Kirschke et al., 2013; Knox et al., 2019; Saunois et al., 2020).

To understand the biogenic $CH_4$ flux potential of the United States of America (US), a multi-scale $CH_4$ observation network focused on $CH_4$ flux rates, processes, and scaling methods is required. When scaling bottom-up measurements to the landscape and regional scale, measurements of $CH_4$ from existing infrastructure tend not to be sufficiently geographically distributed to capture the true spatial variation that is innate to the production and consumption of $CH_4$, and is compounded by large source/sink strengths in small areas (e.g., periodic wetting/drying of seasonal wetlands, saturated soils) (IPCC, 2013; Knox et al., 2019; Thornton et al., 2016) and by very small source/sink strengths in very large areas. In addition, bottom-up biogenic

CH$_4$ process-level estimates have historically been limited to short periods (<1-2 years), are discontinuous (grab sampling), and/or occur only during the growing season at middle and high latitudes (though see Groffman et al.,( 2006) and Arndt et al., (2019) for notable exceptions).

There is a pressing need to assess the capacity of existing infrastructure for current and future applications (Lovett et al., 2007;

Kumar et al., 2016; Jongman et al., 2017; Novick et al., 2018; Villarreal et al., 2018; Chu et al., 2021). The representativeness of research infrastructure is often described in terms of the extent to which the measurements collected at any given location and time represent the conditions at any other location and time, and this is often driven by ecological and climatic conditions (Sulkava et al., 2011; Chu et al., 2021). Representativeness is also measured across landscapes, and studies have evaluated how well tower infrastructure captures the variability observed at specific sites (Chu et al., 2021). These approaches seek to

understand the representativeness of the measurements for a broader landscape, which is critical for upscaling point measurements to regional and global scales. These types of assessments inform the scientific community on how to increase their utility and are often designed to support network design, upscaling, and bias estimation (Chen et al., 2011; Ciais et al., 2014; Jongman et al., 2017; Schimel and Keller, 2015; Villarreal et al., 2018; Kumar et al., 2016). There have been many attempts to assess the representativeness of existing eddy covariance (EC) tower networks for various purposes. To date, no

study has focused on CH$_4$ infrastructure across the US, though many studies have used clustering and ecoregions (Sulkava et al., 2011; Hargrove and Hoffman, 2003), dissimilarity (Yang et al., 2008), and distance measures (Hargrove and Hoffman, 2003; Yang et al., 2008; He et al., 2015; Hoffman et al., 2013) on climatic (Novick et al., 2018) and vegetation type structure and function (Chu et al., 2021) to measure the representativeness of existing research infrastructure. The primary goal of this work is to fill this key knowledge gap by determining the regions where biogenic CH$_4$ infrastructure is needed within the US

in order to constrain both the national and global CH$_4$ budget.

To determine key regions where biogenic CH$_4$ infrastructure is needed within the US we statistically identify gaps in active research infrastructure and evaluate areas where infrastructure can be augmented to include new CH$_4$ measurements. We use a combination of climate data and dominant land cover to guide the scientific community on how we can develop a distributed observational network for the US by leveraging existing infrastructure. While this analysis does not capture the heterogeneity

of the conditions that drive CH$_4$ fluxes at the ecosystem scale, it is designed to evaluate the sampling intensity of research sites at the landscape scale. This coarse resolution influences the capacity to scale ecosystem-level results to the landscape, regional, and national level, which is required for the development of CH$_4$ budgets and emission reduction strategies.

## 2 Methods

### 2.1 Overview

To determine the gaps in physical research infrastructure for ecosystem-scale $CH_4$ fluxes, we need to understand how the current infrastructure is distributed across the US. We focus here on EC flux towers given their capabilities for continuous measurements and use in upscaling flux estimates and are therefore a useful basis for identifying gaps in the current network of $CH_4$ observations. The AmeriFlux network of EC towers was launched in 1996 and grew from about 15 sites in 1997 to more than 110 active sites registered today. It was originally a network of PI-managed sites measuring ecosystem $CO_2$, $H_2O$,

and energy fluxes. The network was established to connect research on field sites representing major climatic and ecological biomes, including tundra, grasslands, savanna, crops, and coniferous, deciduous, and tropical forests. The AmeriFlux community tailored instrumentation to suit each unique ecosystem but now also includes towers that are a part of the standardized network, the National Ecological Observatory Network (NEON). In 2012, the US Department of Energy established the AmeriFlux Management Project (AMP) at Lawrence Berkeley National Laboratory (LBNL) to support the

broad AmeriFlux community and the AmeriFlux sites. The AMP standardizes, post-processes, and makes flux data available to the research community. More recently, flux towers began measuring $CH_4$ in freshwater, coastal, upland, natural, and managed ecosystems. Although we have information on the location of existing EC tower infrastructure that is a part of AmeriFlux (n=223), NEON (n=47), and known, independent PI-managed sites (n=141). We focus this analysis on the towers measuring $CH_4$ (n=100) and we distinguish between towers providing data to AmeriFlux (yes = 49, no = 51) and tower activity

(active = 70; inactive = 30). We understand that additional towers exist within the US, but because these towers are not reporting or providing data to the flux community, we cannot include them in this analysis.

To understand the landscape representativeness across geographic clusters, we measured dissimilarity based on climate and land cover type, as these two factors together are characteristic of regional resource availability and disturbance regimes. First,

we developed a dissimilarity matrix that was condensed down to a two-dimensional ordination to determine regional clusters and calculate cluster dissimilarity for each location within a cluster (Figure 1). It is important to note that a tower should be representative of the ecosystem type and the region where it is stationed (Desai, 2010; Jung et al., 2011; Xiao et al., 2012; Chu et al., 2021); however, the landscape representativeness analysis done here uses a coarser classification of land cover types that are more emblematic of resource availability, and factors that influence how ecosystems function, not the specific

ecosystem type where the tower is situated. Chu et al. (2021) examined the land-cover composition and vegetation characteristics of 214 AmeriFlux tower site footprints. They found that most sites do not represent the dominant land-cover type of the ecosystems they exist within, and when paired with common model-data integration approaches this mismatch introduces biases on the order of 4%–20% for EVI and 6%–20% for the dominant land cover percentage (Chu et al., 2021), making it essential to consider landscape characteristics in the design and evaluation of network infrastructure. Infrastructure

representativeness at the landscape scale is indicative of the capacity to upscale information by climate and the dominant ecosystems of locations within a landscape.

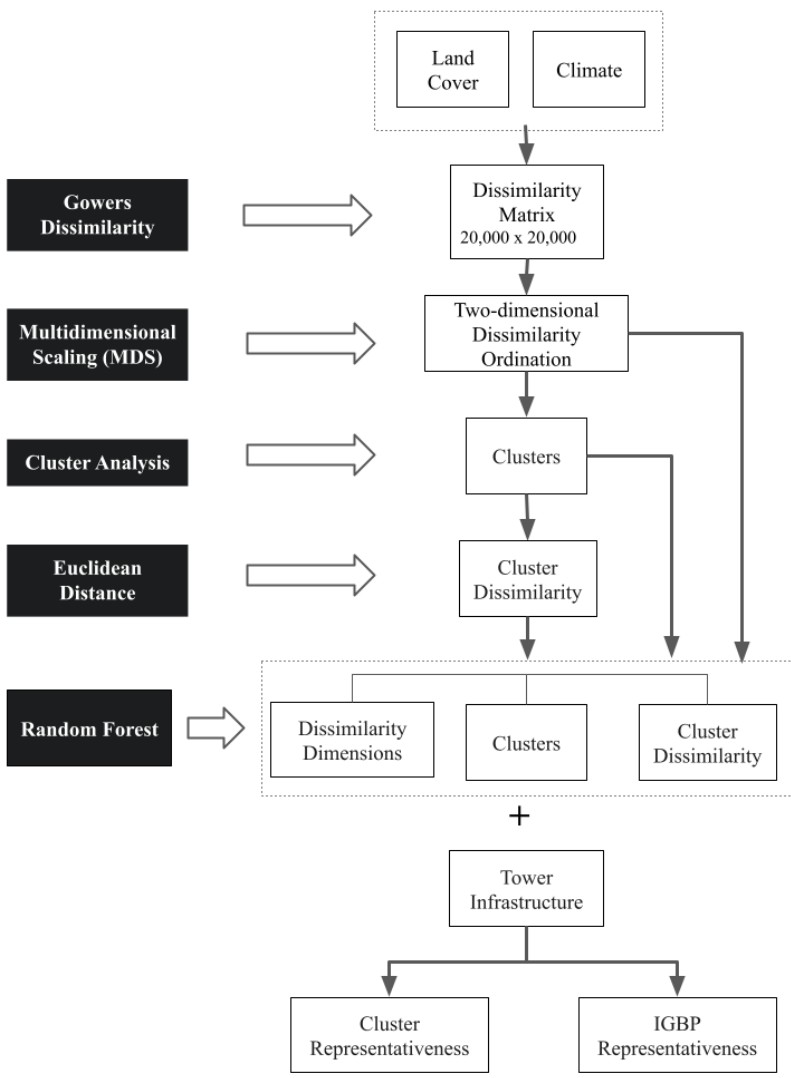

**Figure 1: To determine the gaps in physical research infrastructure for CH4 fluxes we measured landscape cover and climate**
**dissimilarity across the US and evaluated the current distribution of CH4 tower infrastructure.**

## 2.2 Climate and dominant land cover types

We used the National Land Cover Database (NLCD; www.mrlc.gov) to create a land cover layer for the contiguous US (Jin et al., 2019). The NLCD has a 30-m resolution with a 16-class legend based on a modified Anderson Level II classification

system. We reclassified the NLCD into 8 major land cover types (water, developed, barren, forest, scrub, herbaceous, crop, and wetland). Where the NLCD was not available (Alaska, Hawaii, and Puerto Rico), we used the Moderate Resolution Imaging Spectroradiometer (MODIS; 1 km) land cover (type 5 - vegetation functional types) for vegetation functional type (MCD12Q1.006) (Sulla-Menashe and Friedl, 2018), which was also reclassified to the 8 major land cover types (Table 1). The crop land cover type was expanded to non-irrigated and irrigated classes using agricultural information from the US Department of Agriculture's CropScape and Cropland Data layer (Boryan et al., 2011), and the wetland class was expanded using information from the US Fish and Wildlife Service's National Wetland Inventory. Expanded wetland classes were emergent coastal, emergent freshwater, and forest freshwater (Wilen and Bates, 1995). Climate data were obtained from DAYMET (Thornton et al., 2017). We used five climate variables to characterize the climatic conditions across the US: annual mean daily minimum, daily average, and daily maximum temperatures, annual total precipitation, and mean annual daily vapor pressure deficit from 2010-2020. Understanding that these patterns are changing with climate change, we chose a shorter time period than the commonly used 30-year climate normal to better represent current conditions (Bessembinder et al., 2021). Land cover was resampled to match the DAYMET climate data (1-km), and all pre-processing was done in R version 4.0.4 (R Core Team, 2021) with the *raster* package (Hijmans, 2021). This approach allowed us to create a land cover layer of the dominant land cover types at 1 km resolution that was expanded in categories of interest for $CH_4$. The land cover and climate layers were chosen to represent the primary environmental conditions that are often indicative of a combination of resource availability and disturbance regimes. These coarse layers are essential for considering the landscape and large-scale climate effects that can influence how ecosystems within landscapes function. While the available land cover information is appropriate for the coarse, landscape-scale analysis done here, it is important to note that the products used here are not designed to estimate the potential $CH_4$ source/sink status, particularly from the aquatic, wetland, and agricultural land cover types.

**Table 1: Land cover and data sources. The blended land cover product comprises the National Land Cover Database (NLCD) and Moderate Resolution Imaging Spectroradiometer (MODIS). The crop category is enhanced with CropScape and the wetland category with the National Wetland Inventory (NWI) to identify areas dominated by land cover types with additional classes added for types with expected CH$_4$ source potential.**

| Land Cover | Expanded Land Cover | Data Source |
|---|---|---|
| Water | NA | NLCD, MODIS |
| Developed | | |
| Barren | | |
| Forest | | |
| Scrub | | |
| Herbaceous | | |
| Crop | Crops-non irrigated | NLCD, CropScape |
| | Crops-irrigated | NLCD, CropScape |
| Wetlands | Emergent Coastal | NLCD, MODIS, NWI |
| | Emergent Freshwater | |
| | Forested Freshwater | |


**2.3 Measuring landscape dissimilarity across clusters within the US.**

Climate, land cover, and location (latitude/longitude) were used in a multivariate distance analysis (Venables and Ripley, 2002; Ripley, 2007; Cox and Cox, 2008) to measure the dissimilarity across the US (all 50 states & Puerto Rico) at the landscape scale and divide it into ecological clusters. The purpose of this is to identify the interrelatedness of ecological components

within a landscape (Ippoliti et al., 2019). We included location (latitude/longitude) to incorporate the interaction between climate, land cover, and most importantly, seasonality. The US was subsampled because of limitations in the maximum number of points that can be evaluated in the cluster analysis. To measure dissimilarity, we first randomly sampled (n = 20,000 1-km pixels) the US, maintaining the distribution of land cover and climate to define dissimilarity between observations. Although there were more than 8 million 1-km pixels available for the US, there are limits to the number of samples that can be analyzed

by the functions used for the multidimensional scaling (MDS) analysis. We first developed a dissimilarity matrix by calculating Gower dissimilarity (Gower, 1971; Huang, 1997; Podani, 1999; Ahmad and Dey, 2007; Harikumar and Pv, 2015) using the function distmix from the package *kmed* in R. We used Gower dissimilarity because it can handle mixed data types. For each variable type in the data set, the dissimilarity metric that works well for that type is used and scaled to fall between 0 and 1. Then, a linear combination featuring user-specified weights (most simply an average) is calculated to create the final

dissimilarity matrix. This approach measures the dissimilarity for each location within the US using land cover, climate, and

location information (land cover, five climate variables, and location) and creates a dissimilarity matrix (20,000 x 20,000) that indicates dissimilarity for a location to every other location in the US.

Once we created the dissimilarity matrix, we used MDS to generate a two-dimensional ordination showing landscape dissimilarity with the *MASS* package in R (Venables and Ripley, 2002). The MDS makes it possible to evaluate dissimilarity in two dimensions, which is essential to our goal to evaluate representativeness. We used the Kruskal method of non-metric scaling with the IsoMDS function in the *MASS* package (Venables and Ripley, 2002). IsoMDS works best when applied to metric variables (Torgerson, 1958). Torgerson (1958) initially developed this method, which assumes that the data obey distance axioms. It uses eigen decomposition of the dissimilarity to identify major components and axes, and represents any point as a linear combination of dimensions. This is very similar to principal component analysis (PCA) or factor analysis, but it uses the dissimilarity matrix rather than a correlation matrix as input. Furthermore, the included dimensions are the most important dimensions produced, like PCA which is able to identify all of the dimensions that exist in the original data up to N−1, but will retain only the most important ones.

Knowing that regional patterns in climate and land cover will be important for scaling $CH_4$ to the regional and national scale, we divided the US into clusters to evaluate representativeness using the first and second dimension from the MDS. Cluster analysis has been used to assess the spatial representativeness of network infrastructure, and to suggest arrangements of study sites (Sulkava et al., 2011; Kumar et al., 2016). It is an objective method of producing meaningful, mutually exclusive groups based on similarities among entities (Balijepally et al., 2011). This approach is descriptive, a-theoretical, and non-inferential with sound mathematical support (Balijepally et al., 2011). Clustering outcomes are driven by large effect sizes or the accumulation of many smaller effects across features, and are mostly unaffected by differences in covariance structure (Dalmaijer et al., 2020). Sufficient statistical power is achieved with relatively small samples (Dalmaijer et al., 2020), provided cluster separation is sufficient. Traditional notions about statistical power only partially apply to cluster analysis (Dalmaijer et al., 2020). Increasing the number of sample points above a sufficient sample size does not improve power, but effect size is important (Dalmaijer et al., 2020). Clustering is useful when large subgroup separation is expected, and when MDS improves cluster separation (Dalmaijer et al., 2020).

We determined the optimal number of clusters using the library *cluster* and the function pam in R (Reynolds et al., 2006; Schubert and Rousseeuw, 2019, 2021). This approach uses the k-medoids algorithm, which partitions a data set into k groups or clusters and is a robust alternative to k-means clustering (Kaufman and Rousseeuw, 2009). The k-medoid algorithm is less sensitive to noise and outliers, compared to k-means, because it uses medoids as cluster centers. The k-medoids algorithm requires the user to specify k, the number of clusters to be generated. A useful approach to determine the optimal number of clusters is the **silhouette** method. We fit an increasing number of clusters from 2 to 50 to construct a silhouette plot and choose the number of clusters that maximized the average silhouette width (Figure S2).


While useful, there are limitations to cluster analysis that can affect cluster patterns and the stability of clusters. The final cluster solution is dependent upon the clustering variables, the similarity/dissimilarity measure used, the clustering algorithm, and the data used to estimate clusters. Therefore, varying elements of clustering methods can lead to many alternative cluster solutions (Balijepally et al., 2011). Cluster solutions can also be produced in the absence of natural structure in the data, and

there is no statistical basis to reject the null hypothesis that there are no natural groupings in the data (Balijepally et al., 2011). Cluster algorithms also cannot differentiate between relevant versus irrelevant variables. Therefore, only the variables expected to be influential should be used (Balijepally et al., 2011) and should emanate from past research or explicit theory, and be consistent with the objectives of the study.

Due to the limitations of this approach, it is important to validate the cluster solution to ensure its meaningfulness and utility (Punj and Stewart, 1983; Balijepally et al., 2011). Consistency is established by checking the stability of cluster solutions obtained by using multiple algorithms (Punj and Stewart, 1983) or through splitting a sample, analyzing the cluster solutions for the two halves separately, and checking their consistency. After checking for reliability, the validity of a cluster solution is established through external validity and criterion-related validity. External validity ensures that clusters are representative of

the actual population (Cook and Campbell, 1979) and can be verified by clustering on a hold-out sample using the same variables and assessing the similarity of the two solutions. This analysis was repeated 5 times to ensure that the 20,000 pixel subsample would produce similar results in the dimensions and clustering. For simplicity, we show the results of the first analysis, and a comparison of clustering methods and measures of stability are available in the supplement.

To measure dissimilarity across the cluster once defined, each cluster was represented by one of the data points in the cluster named the cluster medoid. The medioid had the lowest average dissimilarity between it and all other objects in the cluster. The medoid can be considered a representative example of the members of that cluster. We calculated the dissimilarity between every location within the cluster to the medoid to create a measure of how different each location was from the medoid condition of each cluster. We utilized the pointDistance function in the *raster* package, which provided a unit-less relative

measure of dissimilarity that was determined by measuring the difference between the first and second dimensions produced by the isoMDS of each point in a cluster to the dimensions of the medoid.

To extrapolate the cluster and dissimilarity layers across the entire US beyond the 20,000-pixel subsample and to show the predictive validity (Kerlinger, 1986) we employed the machine learning algorithm Random Forest (RF) with the package

*randomForest* (Liaw and Wiener, 2002) to model the first and second dimensions using the land cover and climate layers as predictors. We then created a Random Forest model of the cluster layer using the first and second dimension as the explanatory variables. All models were then projected spatially to produce a spatially explicit cluster layer and a dissimilarity layer beyond the 20,000 sample points that were used in the MDS analysis. The RF algorithm was first introduced by (Breiman, 2001), and

uses an ensemble of regression trees to predict target values. In RF, a series of bootstrapped datasets are used to generate independent regression trees; at each node, a random sample of predictor variables is selected for use. The RF prediction is the ensemble of multiple individual trees. We created 500 trees for each year and site, using 80% of the data for model fitting and 20% for model validation. The fit of each RF was evaluated with the out-of-bag mean square error (OOB MSE), and variable importance was computed as the amount of the prediction error increased when a particular predictor was permuted. Initially, 500 RF trees were generated. Overall model fit was evaluated with the average of the 500 OOB MSEs from the final model for each year and site, and variable importance was calculated as the average rank of each predictor variable for the 500 models. This approach allowed us to measure the importance of the original data on the first and second dimensions defined by the MDS and how the MDS leads to cluster and dissimilarity patterns. This step was essential to producing a spatially explicit cluster and dissimilarity layers for the entire US, since the MDS analysis limits the number of observations that can be analyzed. This is also important for evaluating the meaningfulness of the cluster by using the original variables used in the development of the distance matrix to predict clusters.

## 2.4. Measuring the landscape representativeness of research infrastructure.

Representativeness studies discern when, where, and at what frequency networks are measuring ecological processes (Baldocchi et al., 2012; Jongman et al., 2017; Vaughan et al., 2001; Villarreal et al., 2018). To understand the representativeness of current $CH_4$ infrastructure, we defined clusters (Sulkava et al., 2011) and measured the dissimilarity between each location in a cluster to the medoid. We extracted the cluster and dissimilarity for all active tower sites measuring $CH_4$ that were distributed across the US and measured the tower cluster representativeness ($TR_{cluster}$) as the percent overlap between the range of dissimilarity sampled by the infrastructure ($r_{cluster}$) divided by the range of dissimilarity observed in the entire cluster ($r$; Eq. 1).

$$TR_{cluster} = \frac{r_{cluster}}{r} \times 100 \qquad\qquad \text{Eq. 1}$$

We recognize that it is essential to capture the distribution of dissimilarity across an entire cluster to upscale ecosystem measurements. We also report the sampling intensity of the major ecosystem types within the cluster and report the ecosystem representativeness ($TR_{IGBP}$) by the IGBP vegetation types of the towers (Eq. 2).

$$TR_{IGBP} = \frac{r_{IGBP}}{r} \times 100 \qquad\qquad \text{Eq. 2}$$

This approach allows the evaluation of representativeness that is not based on a specific research site, but on the dissimilarity of a location to other locations in the landscape and we use the range, to indicate a capacity to scale within a cluster which is based on both the effects of landscape dominant land cover, climate, and the specific ecosystems measured (IGBP).

 **3 Results**

**3.1 Measuring landscape dissimilarity across clusters within the US**

Land cover, climate, and location were condensed down to two dissimilarity dimensions (Figure 2a).  Both climatic factors and location were the most important variables for determining dimensions and explained 99% of the variance in dimensions (Figure S1). Using the first and second dimensions, the US was divided into 10 clusters (Figure S2) that were distributed across temperature and wetness gradients (Figure 2; Table 2). The coldest zones were in Alaska and included clusters Na and Nb. Cool to temperate clusters in the midwestern and western US include NW, W, and NEa. Temperate clusters extend from the midwestern to the eastern US and include clusters NEb and Ea. Warm regions were distributed across clusters Eb, SW, and SE. Dry clusters (Na, SW, W, & Nb) were distributed across the western US and Alaska, and wet clusters (Ea, Eb, and SE) were in the south-eastern US and Hawaii. Individual clusters represented 7-16% of the US each by area (Table 2) with cluster NW as the largest cluster in the pacific northwest, and the smallest cluster being cluster Nb in the northern half of Alaska.

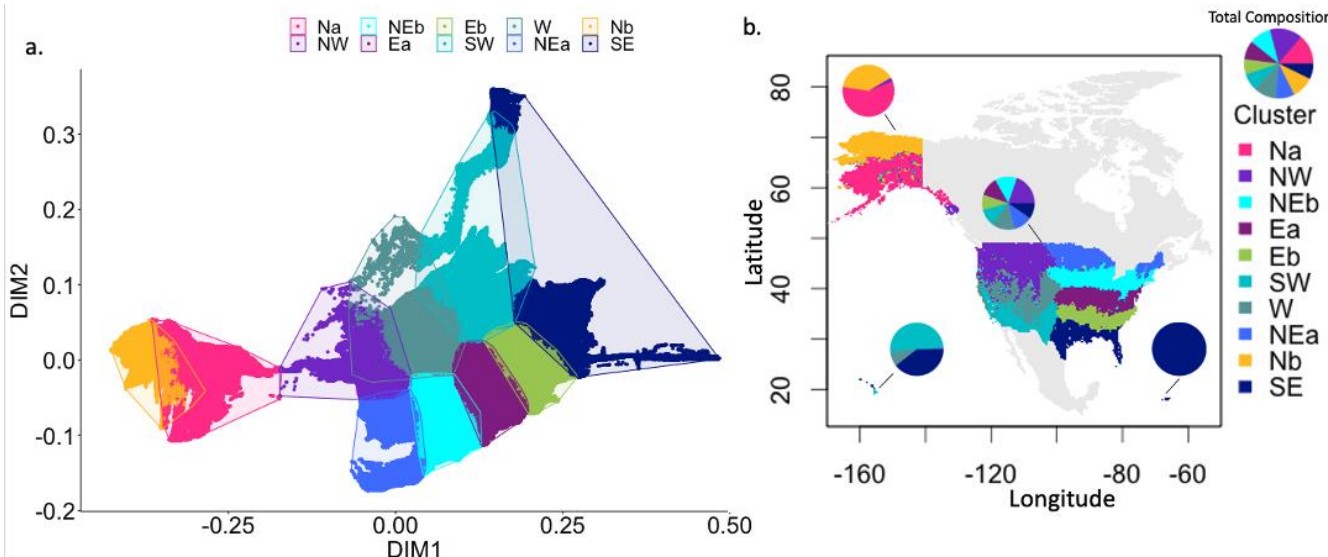

**Figure 2: (a) Multidimensional scaling across the United States (US) produced ten clusters using ecotype (Table1), climate, and location (latitude/longitude). (b) Spatial distribution of the identified clusters.**

Across all clusters, dissimilarity ranged from 0.01 to 0.33 (Figure 3). The mean dissimilarity was 0.04, and most areas within a cluster were less than or equal to the mean. Southern Alaska (cluster Na), Hawaii (clusters SE and Eb), Florida (cluster SE), Puerto Rico (cluster SE), and the northeast (cluster NEa) had greater than average dissimilarity in their respective clusters.

Dominant landscape land cover types also varied across clusters, with forests, scrub, and herbaceous ecosystems dominating clusters (> 20% coverage; Table 2). Although irrigated croplands did not have high coverage rates across any cluster, non-irrigated croplands had high coverage rates in NEb, Ea, and NEb. Wetlands did not have high coverage rates in any cluster.

**Table 2: The land cover and climate of the 10 clusters in the US. Crops were divided into irrigated (CropI) and non-irrigated (CropNI) and wetlands into emergent coastal (WetEC), emergent freshwater (WetEF), and freshwater forest (WetFF). Percent coverage (% Cov) is the percent area occupied by a cluster and R is the range in dissimilarity for each cluster.**

| Cluster | Dominant Landscape Land cover (% Cov) | | | | | | | | Climate | % Cov | R |
|---|---|---|---|---|---|---|---|---|---|---|---|
| | Forest | Scrub | Herb | $Crop_I$ | $Crop_{NI}$ | $Wet_{EC}$ | $We_{EF}$ | $Wet_{FF}$ | | | |
| Na | 27.4 | 39.0 | 3.8 | 0 | 0 | 0.1 | 1.6 | 1.9 | Cold-cool (Dry) | 11 | 0.13 |
| NW | 28.4 | 33.2 | 23.0 | 0 | 6.5 | 0.0 | 0.6 | 0.2 | Cool-Temperate (Mild) | 16 | 0.11 |
| NEb | 24.4 | 0.4 | 9.3 | 0 | 34.2 | 0.0 | 0.7 | 1.9 | Temperate (Mild-Wet) | 10 | 0.06 |
| Ea | 39.0 | 0.9 | 7.7 | 0.0 | 24.3 | 0.3 | 0.2 | 1.4 | Temperate (Wet) | 9 | 0.04 |
| Eb | 37.8 | 3.8 | 9.9 | 0.9 | 14.0 | 0.2 | 0.3 | 6.7 | Warm (Wet) | 8 | 0.04 |
| SW | 2.6 | 58.8 | 17.3 | 0.2 | 7.4 | 0.0 | 0.2 | 0.2 | Warm (dry) | 9 | 0.25 |
| W | 19.4 | 42.6 | 20.8 | 0 | 6.9 | 0.0 | 0.4 | 0.1 | Cool-Temperate (Dry) | 12 | 0.07 |
| NEa | 27.2 | 1.5 | 6.1 | 0 | 23.5 | 0.0 | 2.1 | 7.6 | Cool-Temperate (Mild-Wet) | 9 | 0.08 |
| Nb | 9.0 | 51.7 | 16.0 | 0 | 0 | 0.1 | 2.0 | 0.3 | Cold (Dry) | 7 | 0.11 |
| SE | 19.4 | 19.3 | 8.2 | 0.6 | 7.8 | 1.6 | 2.2 | 7.9 | Hot (Wet) | 9 | 0.31 |

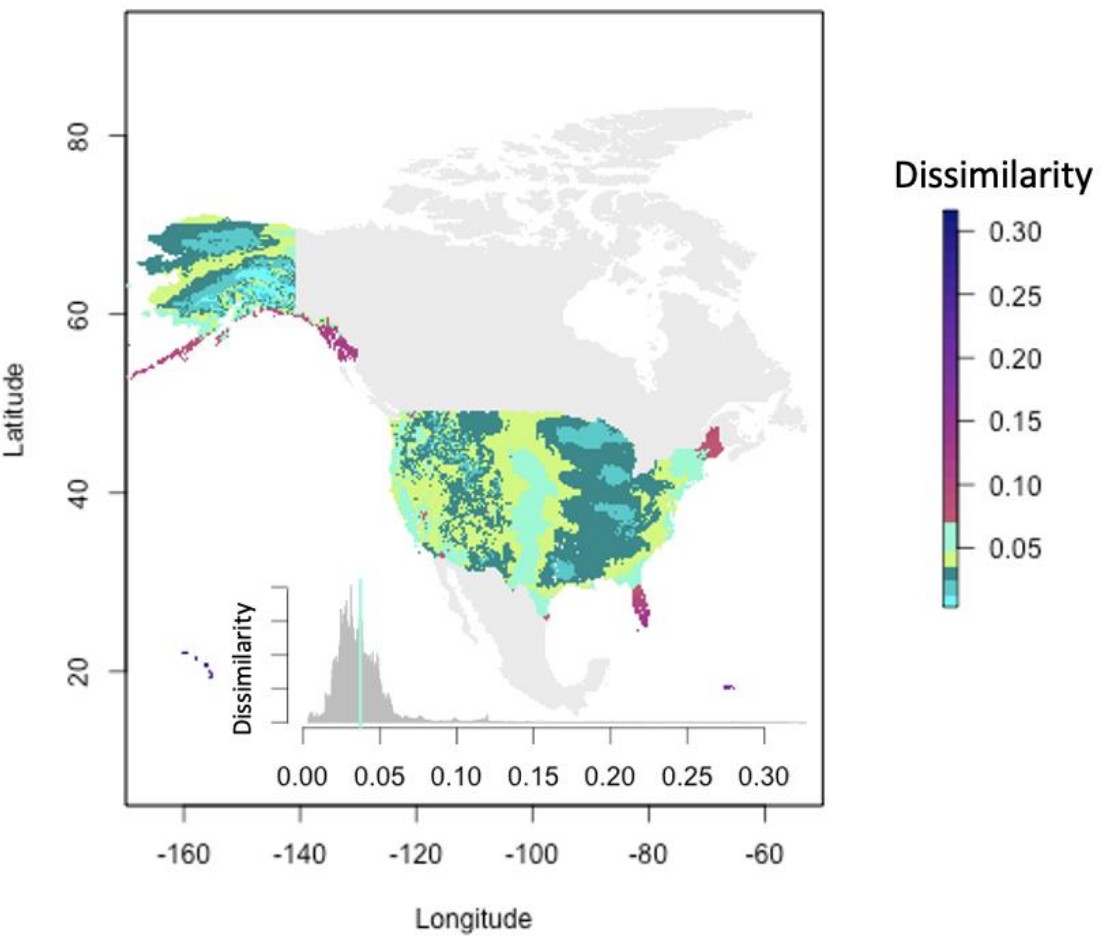

**Figure 3: Cluster dissimilarity for the US. Inset: the distributions of dissimilarity across all clusters shown in a histogram, in which the line denotes the mean dissimilarity across all clusters.**

**3.2 Landscape Representativeness of Existing CH$_4$ Tower Infrastructure**

There were 70 active EC towers measuring CH$_4$ distributed across forest (3 towers), grasslands (4 towers), shrublands (1 tower), agriculture (19 towers), wetlands (37 towers), barren (2 towers), and aquatic (4 towers) IGBP vegetation classes. Less than half of the active towers (43%) were providing data to the community through Ameriflux, limiting the development of CH$_4$-derived products. For this reason, we will first focus this analysis on the active towers providing data to AmeriFlux.

Although CH$_4$ EC tower infrastructure was not a part of a single organized network designed to be representative of the climate, landscape, and dominant IGBP vegetation classes that exist within the US, EC tower infrastructure that was providing data to AmeriFlux was distributed across 8 of the 10 clusters (Table 3), with clusters NW and SE without any active towers providing data to the community. Tower representativeness (TR $_{cluster}$) of clusters ranged from 0 to 88%. The greatest TR$_{cluster}$ was for Eb and NEa and the lowest TR $_{cluster}$ was for NW and SE which had no towers. TR $_{cluster}$ was low (<50%) for most clusters and

high coverage was not associated with a higher frequency of towers. A high TR $_{cluster}$ was found in clusters where towers were

dispersed across IGBP vegetation classes and where towers in wetlands, forests, or the arctic tundra (barren) were distributed across the observed range in the dissimilarity of clusters. Most clusters were substantially under-sampled (Table 3, Figure 4) due to an insufficient number of towers measuring $CH_4$ and poor distribution across the cluster.

The representativeness of IGBP vegetation types within clusters was poor for all vegetation types, excluding forests in the NEa. $TR_{IGBP}$ ranged from 0 to 79% and wetlands were the only IGBP class to be sampled across 8 clusters. Ideally, IGBP classes should be distributed both within and across clusters where the classes exist. There was not a single cluster with towers in all of the IGBP classes (forest, scrub, aquatic ecosystems, crops, wetlands, barren tundra, and grasslands) that are found within that cluster.


**Table 3: The total number of eddy covariance (EC) towers measuring $CH_4$ and providing data to Ameriflux. The tower frequency by dominant landscape type, the total cluster representativeness ($TR_{cluster}$), and cluster representativeness by major ecosystem types are shown ($TR_{IGBP}$). For $TR_{cluster}$ and $TR_{IGBP}$ values of 0.01 were assigned where a single tower was present.**

| Cluster | EC CH$_4$ | Tower Frequency by Dominant Landscape Land Cover | | | | | | | | $TR_{cluster}$ (%) | $TR_{IGBP}$ (%) | | | | | | |
|---|---|---|---|---|---|---|---|---|---|---|---|---|---|---|---|---|---|
| | | Forest | Scrub | Herb | Crop | Wet | Urban | Barren | Water | | Forest | Scrub | AQ | Crop | Wet | Barren | Grass |
| Na | 4 | 2 | 1 | - | - | 1 | - | - | - | 3.0 | 0.01 | 0.01 | - | - | 0.02 | - | - |
| NW | - | - | - | - | - | - | - | - | - | - | - | - | - | - | - | - | - |
| NEb | 2 | 1 | - | - | 1 | - | - | - | - | 19.8 | - | - | - | 0.01 | 0.01 | - | - |
| Ea | 1 | - | - | - | - | 1 | - | - | - | 0.01 | - | - | - | - | 0.01 | - | - |
| Eb | 3 | - | - | - | 1 | 2 | - | - | - | 88 | - | - | - | 0.01 | 42.1 | - | - |
| SW | 7 | - | - | 1 | 3 | 2 | - | 1 | - | 2.0 | - | - | - | 0.14 | 2.0 | - | 0.01 |
| W | 1 | - | - | - | - | - | - | - | 1 | 0.01 | - | - | - | - | 0.01 | - | - |
| NEa | 7 | 4 | - | - | - | 3 | - | - | - | 79.3 | 79.3 | - | 0.02 | - | 13.4 | - | - |
| Nb | 8 | - | 2 | 4 | - | 2 | - | - | - | 21.3 | - | - | 0.01 | - | 21.3 | 6.3 | 0.01 |
| SE | - | - | - | - | - | - | - | - | - | - | - | - | - | - | - | - | - |

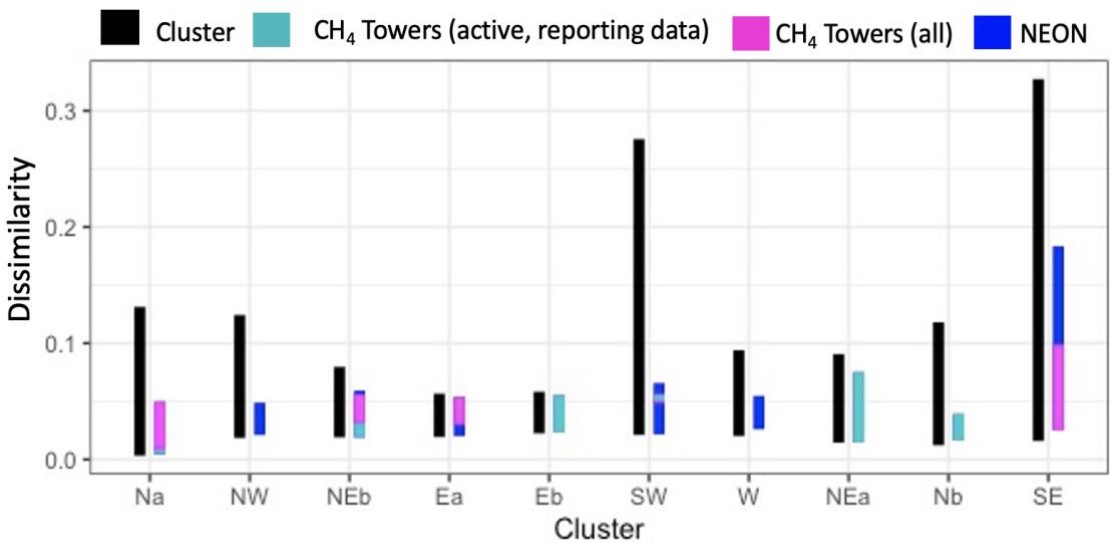


**Figure 4: The range in dissimilarity for clusters (black bar), active CH₄ towers providing CH₄ data to Ameriflux (cyan), all active CH₄ towers (magenta), and for NEON towers (blue). The black lines show the range in dissimilarity observed for a cluster and greater overlap between the cluster range and the tower range is important for landscape representativeness.**

**Table 4: The TR $_{cluster}$ for CH₄ towers that are active and providing data to Ameriflux, the TR $_{cluster}$ for all active CH₄ towers, and**
**the TR $_{cluster}$ for all active towers in addition to NEON towers.**

| Cluster | CH₄ Towers (Data Providing) | CH₄ Towers (All) | NEON Towers |
|---------|------|------|------|
| **Na** | 3.0 | 34.9 | 35.5 |
| **NW** | - | 0.1 | 26.3 |
| **NEb** | 19.8 | 60.6 | 65.9 |
| **Ea** | 0.01 | 63.1 | 89.4 |
| **Eb** | 88.1 | 88.1 | 88.1 |
| **SW** | 2.0 | 3.3 | 17.3 |
| **W** | 0.01 | 0.01 | 38.8 |
| **NEa** | 79.3 | 79.3 | 79.3 |
| **Nb** | 21.3 | 21.3 | 21.3 |
| **SE** | - | 23.6 | 50.8 |

There were important gains in the TR$_{cluster}$ when considering all CH₄ towers regardless of if they were providing data to Ameriflux (Table 4 and Figure 4). The clusters with substantial gains in representativeness (> 10%) include Na, NEb, Ea, and

the SE. The TR $_{cluster}$ of the NW, Ea, SW, W, and the SE would be further enhanced by more than 10% with the addition of

$CH_4$ instrumentation at NEON tower sites.

## 4 Discussion

To determine key regions where biogenic $CH_4$ infrastructure is needed within the US we identified gaps in active research infrastructure. We found that there is an insufficient number of towers measuring $CH_4$, and the distribution of these sites across the range in dissimilarity observed is poor for all clusters. Current EC towers measuring $CH_4$ are in ecosystems known to be

sources of $CH_4$. This is extremely limiting when trying to upscale $CH_4$ fluxes because it leads to a serious bias towards $CH_4$ emissions in model results, and constrains our capacity to appropriately model ecosystems that are $CH_4$ sinks. In this analysis, we include NEON towers because they are purposefully distributed across climate zones and ecosystem types, they provide consistent and standardized measurements, and existing infrastructure at these sites could be quickly adapted to measure $CH_4$ and all data is made publicly available. We understand that for PI managed infrastructure the placement of towers is driven by

the scientific question being asked and research funding priorities (Papale et al., 2015; Mahecha et al., 2017; Villarreal et al., 2018; Knox et al., 2019), but as the number of towers measuring $CH_4$ fluxes continues to grow, consideration for key underrepresented regions where towers are needed or where more efforts are needed for existing but nonreporting towers to contribute to AmeriFlux is of utmost importance. Making all data available must become the standard of the trace gas flux and biogeochemistry communities. Notable infrastructure gaps were in clusters Na, NW, SW, W, Nb, and the SE, and all clusters

require a greater representation of IGBP vegetation types. Our analysis shows that the Na, W, and Nb clusters are the most poorly represented regions, corresponding to Alaska (Na & Nb) and the Rocky Mountains (W), where large elevational changes in the landscape are inherently difficult to capture.

One reason for gaps in $CH_4$ flux tower infrastructure may be the lag in technological capability behind that of $CO_2$ flux

measurements. Methane gas analyzers with sufficient measurement frequency for EC were not common before the late 1990s and early 2000s (Shurpali et al., 1993; Billesbach et al., 1998; Rinne et al., 2007), and the number of commercial options has expanded only more recently (Peltola et al., 2013; Nemitz et al., 2018; Burba et al., 2019; Burba, 2021). Therefore, as the flux tower infrastructure has expanded to measure $CH_4$, decisions on the locations of measurement sites have largely been tied to $CO_2$ and water vapor exchange research (Baldocchi, 2014) and to the availability of suitable infrastructure (McDermitt et al.,

2011), and not necessarily to address $CH_4$ hypotheses. In addition to technological limitations, the environments where we expect $CH_4$ fluxes to be highest complicates considerations for where best to place instrumentation. Large sources of natural biogenic $CH_4$ can sometimes originate from small, heterogeneous components within a landscape, such as patchy wetlands within an otherwise upland forested region, causing the area to be a net source of $CH_4$ (Desai et al., 2015). In contrast, some systems covering large areas that are known to be important $CH_4$ sources, such as arctic tundra ecosystems and shallow lakes

(Wik et al., 2016; Elder et al., 2020), are simply too remote and difficult to instrument. When they are instrumented, towers

are often clustered together regionally, resulting in incremental changes in landscape representativeness. A non-negligible portion of the existing $CH_4$ measurements, including both towers and chambers, are not placed where $CH_4$ sources or sinks are but where the grid power is available to run such measurements. The likely incomplete quantification of $CH_4$ fluxes within heterogeneous sites and the measurement of $CH_4$ fluxes at sites that were established to measure $CO_2$ and energy fluxes, together introduce an inherent source of site-level bias in existing data and our analyses. Hence, we interpret our results as a best-case scenario, as this bias likely would reduce even further our reported degree of representativeness.

Gaps in our US infrastructure and current capability to measure $CH_4$ were most noted when considering only the AmeriFlux sites that provide $CH_4$ data. When evaluating all sites with $CH_4$ infrastructure with the addition of the measurement capability from NEON sites, there were great improvements in landscape representativeness. Still, the largest gaps in infrastructure capability to measure expected $CH_4$ sources were from aquatic sites. These gaps in representation have been noted in other investigations of $CH_4$ flux and budget studies, as a part of larger global $CH_4$ analyses (Saunois et al., 2020) and FLUXNET $CH_4$ flux syntheses (Knox et al., 2019; Delwiche et al., 2021). In fact, the call for more measurements of $CH_4$ from natural sites is not new (Matthews and Fung, 1987; Bartlett and Harriss, 1993; Dlugokencky et al., 2011; Nisbet et al., 2014) and has been advocated as necessary to reduce the uncertainty in $CH_4$ budget estimates from natural ecosystems (Peltola et al., 2019), which is among the largest uncertainty in the global $CH_4$ budget (Saunois et al., 2020). Even areas that have been traditionally thought to have negligible $CH_4$ emission or consumption rates should be monitored because their contribution to $CH_4$ budgets may be significant when considering their large spatial extent. There is also a strong need for a continental $CH_4$ observatory to aid in reducing these uncertainties in the natural $CH_4$ sources and sinks.

A large source of uncertainty in scaling bottom-up $CH_4$ estimates are in the current land use classification (LUC) products (Kirschke et al., 2013; Knox et al., 2019; Saunois et al., 2020), which are not designed to estimate the potential $CH_4$ source/sink status, particularly from aquatic, wetland, and agricultural land cover. Aquatic ecosystems contribute significantly to global $CH_4$ emissions, with emissions increasing from natural to impacted aquatic ecosystems and from coastal to freshwater ecosystems (Rosentreter et al., 2021). Specific ecosystems within the landscape can contribute significantly to landscape-level and regional $CH_4$ source/sink estimates. Aquatic emissions are likely to change in the future due to an increase in urbanization, eutrophication, and positive climate feedbacks (IPCC, 2021). Yet current wetland classifications from land use data products are not suitable to capture these potential changes, or the potential feedbacks they may have on $CH_4$ processes. Wetland classifications are often generalized too broadly in current LUC schemas to accurately scale and predict $CH_4$ flux rates and processes. Small changes in the delineation or characterization of LUC can result in changing the source/sink status of whole regions (Kirschke et al., 2013; Barkley et al., 2017; Knox et al., 2019). For wetlands these include (i) delineation of wetland area, the largest natural $CH_4$ source, especially in regions like Alaska and Florida, (ii) conflation of fluxes from wetlands and fresh waters leading to double counting (Thornton et al., 2016), (iii) classification of saturated soils as non-wetland, possibly missing strong $CH_4$ emission potential. For agricultural lands, we must also consider (iv) deforestation for agricultural use,

which reduces the soil $CH_4$ sink potential (Robertson et al., 2000), or (v) accurate representation of agricultural land $CH_4$ potential when land use includes a complex mixture of ruminants feedlots, manure, and pastures (Lassey, 2008). These potential large sources of uncertainties in biogenic $CH_4$ flux estimates cannot be addressed with the land cover maps currently used to scale $CH_4$ fluxes and the existing distribution of $CH_4$ observation sites (Rosentreter et al., 2021). Hence, if we are to build a US $CH_4$ budget using a scaled-up land use classification scheme (as is done for $CO_2$), we need both better representation

of $CH_4$ measurement sites and better identification and quantification of the $CH_4$ source/sink potential of the land use classes themselves, i.e., specific development of land use classes based on $CH_4$ potential.

Ideally, $CH_4$ measurement infrastructure should have representation of all IGBP vegetation classes within and across clusters, where appropriate, and address the scale of spatial heterogeneity that reduces uncertainty in a national $CH_4$ budget with

confidence limits that can inform both research objectives and mitigation policy. Thus, the incorporation of representative $CH_4$ sources and sink strength is essential to develop national $CH_4$ budgets. Neglecting sinks would further bias models that suggest sources occur where we are confident they do not. Advancing research and our process-level understanding of biogenic $CH_4$, we need to determine the measurement scales to assess the degree of spatial heterogeneity required to reduce uncertainty within and among sites. One means to address the within-site scale of spatial uncertainty is from automated chamber measurements

within flux tower footprints, such as that found in soils, or first- and second-order streams. This would also allow the scientific community to determine the within site $CH_4$ source/sink strength from local (chamber; $<1$ $m^2$), ecosystem (EC flux tower; $\sim 1$ $km^2$) and landscape scales (tower concentrations; $\sim 100s$ $km^2$). At even a larger scale, airborne observations of atmospheric $CH_4$ concentrations can be used to estimate boundary-level surface-atmosphere $CH_4$ fluxes and potentially provide greater spatial coverage than towers (Chang et al., 2014; Zona et al., 2016), and provide a mechanistic link between tower-based and

satellite-derived $CH_4$ estimates.

The rate of global climate change re-enforces the urgency to establish a continental-scale $CH_4$ observatory network that can enable the first national $CH_4$ budget. As it stands, we currently do not know the scale, location, or the magnitude of site-based biogenic $CH_4$ source/sinks to estimate a national budget. For example, we lack the quantitative information about specific

processes (particularly those that are stochastic, *e.g*., temperature sensitivity, susceptibility to drought and flooding, tipping points, etc.) from representative ecosystems that would scale and inform a national $CH_4$ budget. In addition to the current uncertainty in basic ecosystem-level $CH_4$ processes and the way they spatially scale, the backdrop of climate change is also changing the rates of $CH_4$ production and consumption, as well as the $CH_4$ transport pathways. For example, arctic regions are warming faster than most other regions of the world (Serreze and Barry, 2011), turning permafrost into wetlands and changing

traditional $CH_4$ sinks to sources on short time scales (Chadburn et al., 2017; Schaefer, 2019; Yumashev et al., 2019). In temperate areas, higher climate-change-induced variability in precipitation (e.g., higher moisture of upland forested soils, prolonged droughts, etc.) results in a reduction of soil $CH_4$ uptake and a reduced global $CH_4$ sink (Ni and Groffman, 2018). Sea-level rise, which leads to the inundation of coastal regions turning previously dry upland environments into saturated,

anoxic areas, can in some cases increase $CH_4$ production and emission rates (Lu et al., 2018). Hence, we do not have a baseline
US $CH_4$ budget to establish a starting point now and to compare to in the future, and as a baseline to estimate the efficacy of any mitigation decision (policy) made today. As scientists, we are often asked what is the most likely future state of an ecological system, and what is the most likely state of a system given if a decision or action is made today? The current state of $CH_4$ research and its ability to inform these questions are still nascent.

## 5 Conclusions

We used landscape dissimilarity to assess gaps in current $CH_4$ infrastructure at the landscape scale in the US. Evaluating the strengths and limitations of existing measurement infrastructure is critical for strategic augmentation to provide the most valuable information toward reducing uncertainties in future large-scale budget estimations. This analysis complements previous studies based on climatic or vegetation characteristics (Hargrove and Hoffman, 2003; Yang et al., 2008; Villarreal et al., 2018), and identifies regions within the US where gaps are limiting the development of upscaling techniques. To accurately
understand the impact of climate and land cover change on biogenic $CH_4$ emissions, we need a long-term, calibrated, and strategic continental-scale $CH_4$ observatory network. Current gaps in existing measurement infrastructure limit our ability to capture the spatial and temporal variations of biogenic $CH_4$ fluxes and therefore limit our ability to predict future $CH_4$ emissions. Maps of potential $CH_4$ emissions require land cover classification targeted at land cover types like wetlands that are important sources of $CH_4$ to the atmosphere. Aquatic ecosystems like streams and lakes as well as coastal ecosystems are
significant and variable sources of $CH_4$ not well studied on a long-term basis. Through our analysis using climate, land cover, and location variables, we have identified priority areas to enhance research infrastructure to provide a more complete understanding of the $CH_4$ flux potential of ecosystem types in the US. For EC tower locations, dissimilarity coverage was lacking for clusters Na, W, and Nb, and currently clusters Na, W, Eb, and Nb are substantially undersampled. All aquatic sites were undersampled within each cluster. An enhanced network would allow for us to monitor both the response of $CH_4$ fluxes
to climate and land use change as well as to assess the impact of future policy and mitigation strategies.

**Code/Data availability:** The data products produced are available on the Knowledge Network for Biocomplexity (https://knb.ecoinformatics.org; Malone 2021)

**Author contribution**: All authors contributed to conceptualization; SLM, KAA, RC, ARC, JPG, and RKV designed the manuscript; SLM, KAA, JPG, GS, and RKV wrote parts of the manuscript; RKV, SLM and JPG supervised the project/manuscript; SLM, KAA, and YO contributed to data curation and formal analysis, and all authors performed critical reviews.

**Competing interests**: The authors report no competing interests.

**Acknowledgements**

The authors appreciate the substantial contributions of reviewers. The authors would also like to acknowledge the researchers and support staff for their contributions to discussions that led to the idea for this manuscript: Melissa Genazzio, Amy Lafreniere, Kim Nitschke, Lori Bruhwiler, Julia Bryce, Patrick Crill, Amarnath Gupta, Ilya Zaslavsky, Ilkay Altintas, Stephen Hale, Mike Stewart, Michael Thomson and Mark Milutinovich. HWL acknowledges the National Science Foundation (NSF) for ongoing support. NEON is a project sponsored by the NSF and managed under cooperative support agreement (EF-1029808) to Battelle. RKV and AC acknowledge UNH's Collaborative Research Excellence (CoRE) grant. SLM acknowledges support provided by NSF grant #2047687. Any opinions, findings, and conclusions or recommendations expressed in this material are those of the authors and do not necessarily reflect the views of our sponsoring agencies. This is contribution # 1432 from the Institute of Environment at Florida International University.

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
