# Peer review of "Gaps in Network Infrastructure limit our understanding of biogenic methane emissions for the United States"

_Biogeosciences, 2021_

## Author Comment (AC1)

Authors would like to recognize the thoughtful comments provided by the Reviewer which led to several important changes in our approach. We clarified the goals of this study, we focused on tower infrastructure currently measuring $CH_4$, and we better explained how we are measuring representativeness.

**Reviewer 1:**

This study aims to present a representative assessment of network infrastructure for improving our understanding of methane emissions across the US. I respectfully believe that the authors do not present the appropriate analysis for clearly addressing this goal. The authors present a relatively simple way to generate (ecological) clusters and then they list how many sites are in these clusters and evaluate their distance from the medoids. Arguably, the clusters were produced with variables that are relevant for any ecological process and they are not specifically designed to represent drivers of CH4 fluxes (as claimed by the authors). Representativeness is assessed based on the distance of the locations of the current study sites to the medoid, which is arguably a flawed approach as there are underlying assumptions that do not consider spatial heterogeneity of importance for CH4 fluxes. Finally, this study is more associated with a generic network representative analysis of AmeriFlux or GLEON and the authors present a lengthy discussion about limitations of CH4 measurements that are not directly related to the results.

Reviewer 1 highlights the need for greater detail in the approach taken to measure representativeness and our overall goal. In response to this comment we increased the level of detail in the methods section and provide here a summarized response.

The primary goal of this work is to determine key regions where we need $CH_4$ infrastructure within the US. We do this by identifying the gaps in active research infrastructure and evaluating where infrastructure can be adapted to include $CH_4$ measurements. To address this goal, we used a combination of climate data and dominant land cover types to guide the scientific community on how we can develop a distributed observing network for the US and provide a template for the development of similar networks in other regions. We focus here on EC flux towers because they are essential for a bottom-up framework that bridges the gap between point-based chamber measurements and airborne platforms and are therefore a useful basis for identifying gaps in the current network of $CH_4$ observations. Although we initially focused on all tower infrastructure, we now focused on the towers measuring $CH_4$ (n=100) and we distinguished between towers providing data to Ameriflux (yes =49, no = 51) and tower activity (active = 70; inactive = 30).

To understand the landscape representativeness across geographic clusters, we measured dissimilarity (previously called distance to the medoid) based on climate and land cover type. It is important to note that at the ecosystem scale a tower is representative of the ecosystem type and the region where it is stationed (Desai, 2010; Jung et al., 2011; Xiao et al., 2012; Chu et al., 2021); however, the landscape representativeness analysis done here uses a coarser classification of land cover classes that are more emblematic of regional disturbance regimes, resource availability, and factors that influence how ecosystems function, not the specific ecosystem type where the tower is situated. Chu et al., 2021 examined the land-cover composition and

vegetation characteristics of 214 AmeriFlux tower site footprints. They found that most sites do not represent the dominant land-cover type of the landscape and when paired with common model-data integration approaches this mis-match introduces biases on the order of 4%–20% for EVI and 6%–20% for the dominant land cover percentage (Chu et al. 2021), making it essential to consider landscape characteristics in the design and evaluation of network infrastructure. Tower representativeness at the landscape scale is indicative of the capacity to upscale information by climate and the dominant ecosystems of locations within a landscape. We also calculate cluster representativeness by the towers' vegetation type to understand the sampling intensity of each vegetation type within a cluster, which is also an essential component of scaling $CH_4$ fluxes (Knox et al., 2019). In this analysis we used the reported International Geosphere-Biosphere Programme (IGBP) vegetation type classes that are listed for each tower in the Ameriflux data base, where we also checked to ensure towers were currently active and providing data to the network.

**Main comments**

I strongly recommend separating the results from the discussion section. The results are very limited, and the discussion is beyond what is presented. Separating these sections will bring transparency and clarity about what was done and how is proposed to be interpreted.

Authors agree with Reviewer 1 and have separated the results and discussion and increased the level of detail in both sections.

The authors claim that the MDA was used to define the state space into ecological clusters using information that is important for capturing patterns in CH4 (lines 208-225). That said, it is unclear how climate, ecotype and location (lat/long) are specific information relevant for CH4 and not for any other ecological process. It seems to me that this is a generic analysis and then the authors are interpreting this for CH4. I respectfully believe that there is a disconnection between this approach and the overarching goal of the study.

We made changes to the introduction and methods to clarify our objectives. The primary goal of this work is to identify the gaps in active research infrastructure by evaluating the location of ground-based research infrastructure that is and can be adapted to measure $CH_4$. This would provide guidance on how the research community could direct their resources to ensure the US can develop biogenic $CH_4$ budgets by targeting gaps in infrastructure. In addressing this goal, we used a combination of climate data and dominant land cover types along with a multidimensional cluster analysis to guide the scientific community on how we can develop a distributed observing network for the US and provide a template for the development of similar networks in other regions. Below we discuss in detail how we accomplished this goal.

Lines 236-245 – This section of the methods is unclear. Furthermore, I do not think that regions more similar to the medoid are more representative within given cluster, it may only mean that these regions are more similar to what the medoid is and have nothing to do with real representativeness. The authors assume that the medoid is more representative of the cluster but I

think this is a misleading mathematical interpretation that is carried into interpretations of ecoregions and their representativeness. This issue is reflected in how the authors assess representativeness of 411 towers as they compare with their distance to the medoid under the (arguably) incorrect assumption that the closer to the medoid is better and that there is no relevant variability that is important for the representativeness of CH4 fluxes across a specific cluster.

Below we clarify what the medoid is and we adjusted how we measured tower representativeness of clusters in our work. The cluster analysis uses the k-medoids algorithm, which partitions data into k groups or clusters. Each cluster was represented by one of the data points in the cluster named the cluster medoid. The medioid has the lowest average dissimilarity between it and all other objects in the cluster. The medoid can be considered a representative example of the members of that cluster. The k-medoids algorithm requires the user to specify k, the number of clusters to be generated. A useful approach to determine the optimal number of clusters is the **silhouette** method. We fit an increasing number of clusters from 2 to 20 to construct a silhouette plot and choose the number of clusters that maximized the average silhouette width. Once we determined the number of clusters and the medoid of the cluster, we calculated the dissimilarity between every location within the cluster to the medoid to create a measure of how different each location was from the medoid condition of each cluster. We utilized the pointDistance function in the *raster* package, which provided a unit-less relative measure of dissimilarity that was determined by measuring the difference between the first and second dimensions produced by the isoMDS of each point in a cluster to the dimensions of the medioid. This analysis was repeated 10 times to ensure that the 20,000 pixel subsample would produce similar results in the dimensions and clustering. For simplicity, we show the results of the first analysis.

Climate, ecotype, and location (latitude/longitude) were used in a multivariate distance analysis to define the state space of the US (all 50 states & Puerto Rico) at the landscape scale and divide it into ecological clusters using information that is important for capturing continental patterns in biogeochemical cycles. Once we created a dissimilarity matrix, we used multidimensional scaling (MDS) to generate a two-dimensional ordination showing landscape dissimilarity with the *MASS* package in R (Venables WNRipley, 2002). The MDS makes it possible to evaluate dissimilarity in two dimensions, which is essential to our goal to evaluate representativeness. Knowing that regional patterns in climate and land cover will be important for scaling CH4 to the regional and national scale, we divided the US into clusters to evaluate representativeness. This cluster analysis also allowed us to summarize our results within a geographical context, an approach that has been used to delineate spatial sampling domains, to assess the spatial representativeness of networks, and to suggest arrangements of study sites (Sulkava et al. 2011; Kumar et al. 2016). Once clusters were defined we utilized the pointDistance function in the *raster* package, which provided a unit-less relative measure of dissimilarity that was determined by measuring the difference between the first and second dimensions produced by the isoMDS of each point in a cluster to the dimensions of the medioid.

To understand representative of current $CH_4$ infrastructure, we defined clusters (Sulkava et al. 2011) and measured the dissimilarity between each location in a cluster to the medoid. We extracted the cluster and dissimilarity for all active tower sites measuring $CH_4$ that were distributed across the US and measured the tower cluster representativeness as the percent overlap between the range of dissimilarity sampled by the infrastructure ($r_{cluster}$) divided by the range of dissimilarity observed in the entire cluster ($r$; Eq. 1).

$TR_{cluster} = (r_{cluster}/r)*100$                                                           Eq. 1

We recognize that it is essential to capture the distribution of dissimilarity across an entire cluster to upscale ecosystem measurements. We also report the sampling intensity of the major ecosystem types within the cluster and report the ecosystem representativeness ($R_{IGBP}$) by the IGBP vegetation types of the towers (Eq. 2).

$TR_{IGBP} = (r_{IGBP}/r)*100$                                                              Eq. 2

This approach allows the evaluation of representativeness that is not based on a specific research site, but on the dissimilarity of a location to other locations in the landscape and we use the range, which is indicative of a capacity to scale within a cluster.

Please note, previously we measure representativeness by looking at the frequency of towers and comparing the distribution of dissimilarity (distance to the medoid) of tower locations to the distribution for the entire cluster. We have updated this approach by focusing on the range in dissimilarity.

Figure 2 – Are these regressions statistically significant? I doubt that that Fig2a is significant and Fig2b needs to be tested. If there is no statistical significance, please remove the line as it is a misleading graphic.

We originally included the regression lines to show (a) the lack of trend and (b) the trend between the % coverage of clusters and the frequenquency of towers within clusters for all tower infrastructure and infrastructure measuring $CH_4$. We removed this figure, as we no longer include all EC towers in this analysis and just focus on active $CH_4$ infrastructure.

Lines 298-307- Are these 411 towers actually active? It will be important to disclaim how many are active or if this is a network analysis of historical sites. Furthermore, not all sites may be relevant or would have equal weights for our understanding of CH4 fluxes. Sites were originally installed to measured CO2 and H2O fluxes but arguably they may not be relevant for regional CH4 fluxes. This question is not addressed in this study but is critical for assessment of the representativeness of a CH4 network.

We initially included all tower infrastructure to garner widespread support for instrumenting all towers. In response to Reviewer comments we focus this analysis on the towers measuring $CH_4$ (n=100) only and we distinguish between towers providing data to Ameriflux (yes =49, no = 51)

and tower activity (active = 70; inactive = 30).  There were 70 active EC towers measuring $CH_4$ distributed across forest (3 towers), grasslands (4 towers), shrublands (1),  agriculture (19 towers), wetland (37 towers), barren (2 towers), and aquatic (4 towers) IGBP vegetation classes. Less than half of the active towers (43%) were providing data to the community through Ameriflux, limiting the development of $CH_4$ derived products. For this reason, we will first focus this analysis on the active towers providing data to Ameriflux. Although $CH_4$ EC tower infrastructure was not a part of a single organized network designed to be representative of the climate, landscape, and dominant IGBP vegetation classes that exist within the US, EC tower infrastructure that was providing data to Ameriflux was distributed across 8 of the 10 clusters (Table 3), with clusters NW and SE without any active towers providing data to the community. Tower representativeness of clusters range from 0 to 88%. The greatest TR cluster was for Eb and NEa and the lowest TR cluster was for NW and SE which had no towers.  TR cluster  was poor (<50%) for most clusters and high coverage was not associated with a higher frequency of towers. A high TR cluster representativeness was found in clusters where towers were dispersed across IGBP vegetation classes and where towers in wetlands, forests, or the arctic tundra (barren) were distributed across the state space of the cluster. Most clusters were substantially under-sampled (Table 3, Figure 4c) due to an insufficient number of towers measuring $CH_4$ and poor distribution across the cluster.

[Figure]

**Figure 4. The range of dissimilarity for clusters, active $CH_4$ towers providing $CH_4$ data to Ameriflux, all active $CH_4$ towers, and for NEON towers.**

**Table 4. The R cluster for $CH_4$ towers that are active and providing data to Ameriflux, the R cluster for all active $CH_4$ towers and the R cluster for all active towers in addition to NEON towers.**

| Cluster | CH$_4$ Towers (Data Providing) | CH$_4$ Towers (All) | NEON Towers |
|---|---|---|---|
| Na | 3.0 | 34.9 | 35.5 |
| NW | - | 0.1 | 26.3 |
| NEb | 19.8 | 60.6 | 65.9 |
| Ea | 0.01 | 63.1 | 89.4 |
| Eb | 88.1 | 88.1 | 88.1 |
| SW | 2.0 | 3.3 | 17.3 |
| W | 0.01 | 0.01 | 38.8 |
| NEa | 79.3 | 79.3 | 79.3 |
| Nb | 21.3 | 21.3 | 21.3 |
| SE | - | 23.6 | 50.8 |

There were important gains in TR $_{cluster}$ when considering all CH$_4$ towers regardless of if they were providing data to Ameriflux (Table 4 and Figure 4). The clusters with substantial gains in representativeness (> 10%) include Na, NEb, Ea, and the SE.  The TR $_{cluster}$ of the NW, Ea, SW, W, and the SE would be enhanced by more than 10% with the addition of CH$_4$ instrumentation at NEON sites.

Lines 305-307 – I respectfully do not think that assessing the distance to a medoid is a good assessment of representativeness. If so, then we should place a few towers in these medoids and we will have a perfect representativeness for each cluster. We also know that clusters have similar ecological characteristics but there is much more diversity and heterogeneity that is not captured within a medoid. The last sentence of this paragraph is misleading as it implies that towers must be placed in the medoids that were calculated with generic variables that arguably are not specific for CH4 fluxes (as they are generic for any ecological process). Similar arguments can be done for the analysis and discussion presented in section 3.3. I respectfully do not think this is the proper way to assess representativeness of places where we need to be measuring CH4 fluxes.

We agree that although the medoid would be a good area to place towers within, to really enhance cluster representativeness towers need to be placed across the range of dissimilarity observed. We made changes to the text and included  new estimates of tower representativeness to the cluster and for IGBP representativeness within a cluster.

Although CH$_4$ EC tower infrastructure was not a part of a single organized network designed to be representative of the climate, landscape, and dominant IGBP vegetation classes that exist within the US, EC tower infrastructure that was providing data to Ameriflux was distributed across 8 of the 10 clusters (Table 3), with clusters NW and SE without any active towers providing data to the community. Tower representativeness of clusters range from 0 to 88%. The greatest TR $_{cluster}$ was for Eb and NEa and the lowest TR $_{cluster}$ was for NW and SE which had no towers.  TR $_{cluster}$  was poor (<50%) for most clusters and high coverage was not associated with a

higher frequency of towers. A high TR $_{cluster}$ representativeness was found in clusters where towers were dispersed across IGBP vegetation classes and where towers in wetlands, forests, or the arctic tundra (barren) were distributed across the state space of the cluster. Most clusters were substantially under-sampled (Table 3) due to an insufficient number of towers measuring $CH_4$ and poor distribution across the cluster.

The representativeness of IGBP vegetation types within clusters was poor for all vegetation types, excluding forests in the NEa. TR $_{IGBP}$ ranged from 0 to 79% and wetlands were the only IGBP class to be sampled across 8 clusters. Ideally, IGBP classes should be distributed both within and across clusters but there was not a single cluster with all 7 IGBP classes (forest, scrub, aquatic ecosystems, crops, wetlands, barren tundra, and grasslands).

**Table 3. The total number of eddy covariance (EC) towers measuring $CH_4$ and providing data to Ameriflux. The tower frequency by dominant landscape type, the total cluster representativeness, and cluster representativeness by major ecosystem types are shown. For R $_{cluster}$ and R $_{ecosystem}$ values of 0.01 were assigned where a single tower is present.**

| Cluster | EC CH$_4$ | Tower Frequency by Dominant Landscape Ecotype | | | | | | | | TR $_{cluster}$ (%) | TR$_{IGBP}$ (%) | | | | | | |
|---|---|---|---|---|---|---|---|---|---|---|---|---|---|---|---|---|---|
| | | Forest | Scrub | Herb | Crop | Wet | Urban | Barren | AQ | | Forest | Scrub | AQ | Crop | Wet | Barren | Grass |
| Na | 4 | 2 | 1 | - | - | 1 | - | - | - | 3.0 | 0.01 | 0.01 | - | - | 0.02 | - | - |
| NW | - | - | - | - | - | - | - | - | - | - | - | - | - | - | - | - | - |
| NEb | 2 | 1 | - | - | 1 | - | - | - | - | 19.8 | - | - | - | 0.01 | 0.01 | - | - |
| Ea | 1 | - | - | - | - | 1 | - | - | - | 0.01 | - | - | - | - | 0.01 | - | - |
| Eb | 3 | - | - | - | 1 | 2 | - | - | - | 88 | - | - | - | 0.01 | 42.1 | - | - |
| SW | 7 | - | - | 1 | 3 | 2 | - | 1 | - | 2.0 | - | - | - | 0.14 | 2.0 | - | 0.01 |
| W | 1 | - | - | - | - | - | - | - | 1 | 0.01 | - | - | - | - | 0.01 | - | - |
| NEa | 7 | 4 | - | - | - | 3 | - | - | - | 79.3 | 79.3 | - | 0.02 | - | 13.4 | - | - |
| Nb | 8 | - | 2 | 4 | - | 2 | - | - | - | 21.3 | - | - | 0.01 | - | 21.3 | 6.3 | 0.01 |
| SE | - | - | - | - | - | - | - | - | - | - | - | - | - | - | - | - | - |

Lines 381-390 – The authors assume that uncertainty is associated to poor data coverage, but this is never assessed. This paragraph essentially calls for more locations for measurements away from the medoid which will imply that representativeness (based on the method proposed by the authors) will be lower, as sites are away from the medoid. This is confusing and I strongly encourage the authors to revise the methods and the interpretation of the results.

We agree that changes in the methods and results are warranted to focus on the goals of this study and to clarify how representativeness is quantified. We also put the methods used in this study in the context of  those used in other efforts to assess the representativeness of networks

and infrastructure for current and future applications (Kumar et al. 2016; Lovett et al. 2007; Jongman et al. 2017; Villarreal et al. 2018; Chu et al. 2021; Novick et al. 2018). Representativeness studies discern when, where, and at what frequency networks are measuring ecological processes (Baldocchi et al. 2012; Jongman et al. 2017; Vaughan et al. 2001; Villarreal et al. 2018). Representativeness of research infrastructure is often described in terms of the extent to which the measurements collected at any given location and time represent the conditions at any other location and time, and this is often driven by ecological and climatic conditions (Sulkava et al. 2011; Chu et al. 2021). Representativeness is also measured across a landscape and studies have evaluated how tower infrastructure captures the variability observed within landscapes. All of these approaches are with the goal of understanding the representativeness of the measurements for a broader landscape, which is critical for upscaling point measurements to regional and global scales. Assessments inform the scientific community on how to increase their utility and are often designed to support network design, upscaling, and bias estimation (Chen et al. 2011; Ciais et al. 2014; Jongman et al. 2017; Schimel and Keller 2015; Villarreal et al. 2018; Kumar et al. 2016). There have been many attempts to assess the representativeness of existing flux tower networks for various purposes. To date, no study has focused on $CH_4$ infrastructure across the US, though many studies have used clustering and ecoregions (Sulkava et al. 2011; Hargrove et al. 2003), dissimilarity (Yang et al. 2008), and distance measures (Hargrove et al. 2003; Yang et al. 2008; He et al. 2015; Hoffman et al. 2013) on climatic (Novick et al. 2018) and vegetation type structure and function (Chu et al. 2021).

To understand representative of current $CH_4$ infrastructure, we defined clusters (Sulkava et al. 2011) and measured the dissimilarity between each location in a cluster to the medoid. We extracted the cluster and dissimilarity for all active tower sites measuring $CH_4$ that were distributed across the US and measured the tower cluster representativeness as the percent overlap between the range of dissimilarity sampled by the infrastructure (r $_{cluster}$) divided by the range of dissimilarity observed in the entire cluster (*r*; Eq. 1).

This approach allows us to identify key regions where we need $CH_4$ infrastructure within the US. We agree that this analysis does not capture the heterogeneity of the conditions that drive $CH_4$ fluxes at the ecosystem scale. It is designed to evaluate the sampling intensity of research sites at the landscape scale. In the design of a network, this course resolution influences the capacity to scale ecosystem level results to the landscape, region, and to the national level, which is required for the development of budgets and emission strategies.

Lines 293-402 – This is a similar paragraph where the authors discuss about uncertainty from a narrative, but this was never quantified in the formal representativeness analysis presented in this study. This paragraph and most of the discussion section is an expert opinion and is not directly related to the analyses presented.

Thank you for this comment, we made changes to the text throughout to better connect the discussion to the goals of this work. While we are interested in reducing uncertainties in CH4 budgets and models, we refocused the discussion on evaluating the strengths and limitations of existing measurement infrastructure and the critical need for strategic augmentation to provide the most valuable information toward reducing uncertainties in future large-scale budget

estimations. Our analysis complements previous studies based on climatic or vegetation characteristics (Hargrove et al. 2003; Yang et al. 2008; Villarreal et al. 2018), and identifies regions within the US where gaps are limiting the development of upscaling techniques. To accurately understand the impact of climate and land cover change on biogenic $CH_4$ emissions, we need a long-term, calibrated, and strategic continental-scale $CH_4$ observatory network. Current gaps in existing measurement infrastructure limit our ability to capture the spatial and temporal variation of biogenic $CH_4$ fluxes and therefore limit our ability to predict future $CH_4$ emissions. Maps of potential $CH_4$ emissions require land cover classification targeted at land cover types like wetlands that are important sources of $CH_4$ to the atmosphere. Aquatic ecosystems like streams and lakes as well as coastal ecosystems are significant and variable sources of $CH_4$ not well studied on a long-term basis. Through our analysis using climate, land cover, and location variables, we have identified priority areas to enhance research infrastructure to provide a more complete understanding of the $CH_4$ flux potential of ecosystem types in the US. For EC tower locations, dissimilarity coverage was lacking for clusters Na, W, and Nb, and currently clusters Na, W, Eb, and Nb are substantially under sampled. All aquatic sites are under sampled within each cluster. An enhanced network would allow for us to monitor both the response of $CH_4$ fluxes to climate and land use change as well as the impact of future policy interventions and mitigation strategies.

There are three related studies that assess the representativeness of the AmeriFlux network that may be of interest for the authors.

Chu, H., X. Luo, Z. Ouyang, W. S. Chan, S. Dengel, S. C. Biraud, M. S. Torn, S. Metzger, J. Kumar, M. A. Arain, T. J. Arkebauer, D. Baldocchi, C. Bernacchi, D. Billesbach, T. A. Black, P. D. Blanken, G. Bohrer, R. Bracho, S. Brown, N. A. Brunsell, J. Chen, X. Chen, K. Clark, A. R. Desai, T. Duman, D. Durden, S. Fares, I. Forbrich, J. A. Gamon, C. M. Gough, T. Griffis, M. Helbig, D. Hollinger, E. Humphreys, H. Ikawa, H. Iwata, Y. Ju, J. F. Knowles, S. H. Knox, H. Kobayashi, T. Kolb, B. Law, X. Lee, M. Litvak, H. Liu, J. W. Munger, A. Noormets, K. Novick, S. F. Oberbauer, W. Oechel, P. Oikawa, S. A. Papuga, E. Pendall, P. Prajapati, J. Prueger, W. L. Quinton, A. D. Richardson, E. S. Russell, R. L. Scott, G. Starr, R. Staebler, P. C. Stoy, E. Stuart-Haëntjens, O. Sonnentag, R. C. Sullivan, A. Suyker, M. Ueyama, R. Vargas, J. D. Wood, and D. Zona. 2021. Representativeness of Eddy-Covariance flux footprints for areas surrounding AmeriFlux sites. Agricultural and Forest Meteorology 301-302:108350.

Thank you for this comment. We agree that the work of Chu et al., 2021 is relevant to our goals and provides an important analysis of EC tower footprint-to-target-area mismatch. He showed that few eddy-covariance sites are located in a truly homogeneous landscapes when considering climate and land cover characteristics . Mis-match is limiting model-data integration approaches and introducing biases on the order of 4%–20% for EVI and 6%–20% for the dominant land cover percentage.

We considered the results of their analysis, which support the evaluation of dominant land cover types in the evaluation of representativeness. Chu et al., 2021 chose land cover type as the categorical characteristic because it is commonly used in modeling and upscaling studies. The land cover products used in this study include the 2001–2016 United States National Land Cover Dataset products (NLCD; https://www.mrlc.gov/)and 2010 Land Cover of Canada (https://open.canada.ca/). We used similar products as Chu etal., 2021, improved to provide more detail on aquatic systems due to their importance for $CH_4$. It is important that towers are representative of the landscapes they exist within, and this work proposed a simple representativeness index based on their evaluations that can be used as a guide to identify site-periods suitable for specific applications and to provide general guidance for data use.

Novick, K. A., J. A. Biederman, A. R. Desai, M. E. Litvak, D. J. P. Moore, R. L. Scott, and M. S. Torn. 2018. The AmeriFlux network: A coalition of the willing. Agricultural and Forest Meteorology 249:444–456.

Novick et al., 2018 is another great and relevant synthesis study that laid the foundation for our work. Novick et al. 2018, discusses representativeness in reference to the climate (MAT and MAP) of towers, noting the degree of overlap in network infrastructure. This overlap makes it possible to subsample from the AmeriFlux database to form site-clusters that experience similar climate conditions but different land cover types, enabling the disentangling of effects of climate and vegetation on fluxes. The dissimilarity measure across clusters is used here to measure the variation across clusters and we are interested in current $CH_4$ infrastructure in this landscape.

Villarreal, S., M. Guevara, D. Alcaraz-Segura, N. A. Brunsell, D. Hayes, H. W. Loescher, and R. Vargas. 2018. Ecosystem functional diversity and the representativeness of environmental networks across the conterminous United States. Agricultural and Forest Meteorology 262:423–43while we used the dominant landscape lcc.

Villarreal et al., 2018 assess the representativeness of AmeriFlux and NEON based on ecosystem functional diversity characterized by 64 EFT categories across CONUS. Their EFT analysis defined the prominent EFT for a location (EFTmode) and measured representativeness based on a) the number of different EFT categories (EFTmode) represented by each network, b) representativeness of the EFT inter-annual variability (EFTint; number of unique EFTs within each pixel during years 2001–2014), and c) the spatial representation of EFTmode and EFTint based on a maximum entropy approach (i.e., spatial functional heterogeneity).

We included these studies and more to put this work in the broader context of representative studies. "There is a pressing need to design different scientific approaches to assess the representativeness of networks and infrastructure for current and future applications (Kumar et al. 2016; Lovett et al. 2007; Jongman et al. 2017; Villarreal et al. 2018; Chu et al. 2021; Novick

et al. 2018). Representativeness studies discern when, where, and at what frequency networks are measuring ecological processes (Baldocchi et al. 2012; Jongman et al. 2017; Vaughan et al. 2001; Villarreal et al. 2018). Representativeness of research infrastructure is often described in terms of the extent to which the measurements collected at any given location and time represent the conditions at any other location and time, and this is often driven by ecological and climatic conditions (Sulkava et al. 2011; Chu et al. 2021). Representativeness is also measured across a landscape and studies have evaluated how tower infrastructure captures the variability observed within landscapes. All of these approaches are with the goal of understanding the representativeness of the measurements for a broader landscape, which is critical for upscaling point measurements to regional and global scales. Assessments inform the scientific community on how to increase their utility and are often designed to support network design, upscaling, and bias estimation (Chen et al. 2011; Ciais et al. 2014; Jongman et al. 2017; Schimel and Keller 2015; Villarreal et al. 2018; Kumar et al. 2016). There have been many attempts to assess the representativeness of existing flux tower networks for various purposes. To date, no study has focused on $CH_4$ infrastructure across the US, though many studies have used clustering and ecoregions (Sulkava et al. 2011; Hargrove et al. 2003), dissimilarity (Yang et al. 2008), and distance measures (Hargrove et al. 2003; Yang et al. 2008; He et al. 2015; Hoffman et al. 2013) on climatic (Novick et al. 2018) and vegetation type structure and function (Chu et al. 2021)."

---

## Author Comment (AC2)

Authors would like to recognize the thoughtful comments provided by the Reviewer which led to several important changes in our approach. We clarified the goals of this study, we focused on tower infrastructure currently measuring $CH_4$, and we better explained how we are measuring representativeness.

**Reviewer 2**

This study by Malone et. al. presents an representativeness analysis of current Eddy Covariance (EC) CH4 observation sites to understand gaps in data collection and develop guidance for new research infrastructure to reduce the uncertainties in US CH4 budget. It applies a multidimensional cluster analysis based approach to develop and assess CH4 observing network for US.

Study outlines well the limitations and uncertainties associates with current EC CH4 network in US and takes on an important research to better understand its representation. However, there are number of limitations in the data and methodoloy applied in the study, presented results does not support the conclusions and overall it does not provide much improvements over limitations in current literature that manuscript describes in introduction.

In response to Reviewer 2, we made important clarifications in the methods used and changed the way we discuss representativeness. We provide details below.

Comments on methods:

Method section is brief, lack necessary description and is at time diffcult to understand and follow.

- While cluster analysis method in itself is sound, its unclear if and how the choice of land cover and climate layers chosen to represent primary environmental conditions represent ecosystem scale CH4 fluxes. And if and how they represent the conditions for aquatic sites?

We used a combination of climate data and dominant land cover types to guide the scientific community on how we can develop a distributed observing network for the US and provide a template for the development of similar networks in other regions.  We focus here on EC flux towers because they are essential for a bottom-up framework that bridges the gap between point-based chamber measurements and airborne platforms and are therefore a useful basis for identifying gaps in the current network of $CH_4$ observations. To understand the landscape representativeness across geographic clusters, we measured dissimilarity based on climate and land cover type, as these two factors together are characteristic of regional resource availability and disturbance regimes. It is important to note that a tower is representative of the ecosystem type and the region where it is stationed (Desai, 2010; Jung et al., 2011; Xiao et al., 2012; Chu et al., 2021); however, the landscape representativeness analysis done here uses a coarser classification of ecosystem types that are more emblematic of regional disturbance regimes, resource availability, and factors that influence how ecosystems function, not the specific

ecosystem type where the tower is situated. Chu et al., 2021 examined the land-cover composition and vegetation characteristics of 214 AmeriFlux tower site footprints. They found that most sites do not represent the dominant land-cover type of the ecosystems they exists within and when paired with common model-data integration approaches this mis-match introduces biases on the order of 4%–20% for EVI and 6%–20% for the dominant land cover percentage (Chu et al. 2021), making it essential to consider landscape characteristics in the design and evaluation of network infrastructure. Tower representativeness at the landscape scale is indicative of the capacity to upscale information by climate and the dominant ecosystems of locations within a landscape. We also calculate regional representativeness by the towers' vegetation type to understand the sampling intensity of each vegetation type within a cluster, which is also an essential component of scaling $CH_4$ fluxes (Knox et al., 2019). In this analysis we used the reported International Geosphere-Biosphere Programme (IGBP) vegetation type that is listed for each tower in the Ameriflux data base, where we also checked to ensure towers were currently active and providing data to the network.

- Authors highlight on Line 75-79 the limitations of existing land use products to identify CH4 source/sink, wetland classifications not suitable to scale and predict CH4 flux rates and processes. BUT they choose to use existing NLCD reclassified to 8 classes and reduced 1km resolution, thus effectively reducing the fidelity of the landcover data. Did do however improve the representation of irrigated vs non-irrigated crop ecotypes, and wetland classes. However, no quantitative analysis has been presented to demonstrate improvements their landcover scheme provide over the existing landcover.

Thank you for this comment. We moved the information on the limitation of current land cover products from the introduction to the discussion. We did this because having it in the introduction combined with the changes we made in the product gave the impression that the changes we made improved on the current product in a way that would reduce uncertainties. In fact that is not the case, the same uncertainties exist. The product we developed only allowed us to distinguish a few wetland and aquatic classes from each other.

- Line 198-200 states "The land cover and climate layers were chosen to represent the primary environmental conditions that are often indicative of a combination of resource availability and disturbance regimes." What resource availability refers to in this particular context? Also, which layer, if any, captures the disturbance regimes and what type of disturbances?

Land cover composition is a response to climate and disturbance regimes. Climatic factors influence resource availability (light, water, temperature) and although it varies geographically, disturbance is tightly linked to land cover types and cover classes have characteristic disturbances that influence the composition of classes within a landscape (Hermosilla et al. 2018). Studies have shown that either changes in climate or disturbance can lead to shifts in landscape composition (Sharp and Angelini 2019; Wang et al. 2020). For example, changes in the plant functional types across Arctic–Boreal ecosystems have been linked to wildfires and

climate-driven expansion of woody shrubs (Wang et al. 2020). The interaction of climate and disturbance both influence landscape structure and develop spatial patterns of many ecosystem processes on the landscape (Turner 2010).

- Section 2.3 Defining the state space of the US

  - Multidimensional scaling (MDS) was performed to condense ecotype, climate and location variables to two dimension. But unlike climate and location, ecotype is a categorical variable and how was a categorical value handled in the MDS scheme. Why were they condensed down to two dimension, aside from R/MASS implementation's default?

[revised manuscript text omitted]

and dissimilarity layers for the entire US, since the MDS analysis limits the number of observations that can be analyzed.  We added more detail to the text and the results of the random forest model and the importance of the original data in explaining clustering and the first and second MDS dimensions.

- ○ Lines 218-220 states "We measured the correlation between the ecotypes, climate layers, and locations (latitude/longitude) using the envfit function in the library vegan in R (Oksanen, 2016)."  What was the purpose of these correlations? And again how do you calculated correlation against categorical ecotype variable? How meaningful is it to calculated the correlation of climate or ecotype to location lat/lon? These choice of methods need some clarification beyond reference to R packages.

This was originally included to explain the main determinants of the MDS and clustering. We removed it from the analysis. The results are redundant with the inclusion of the random  forest variable importance plots, which will be included in the supplement.

- ○ Lines 219-223 "This was followed by a cluster analysis to determine the optimal number of clusters using the library cluster in R, which partitions data around medoids (PAM algorithm), using the Gower dissimilarity matrix (Gower, 1971; Huang, 1997; Podani, 1999; Ahmad and Dey, 2007; Harikumar and Pv, 2015)." Its unclear if clustering was done on orignal set of ecotype, climate and location variables or two MDS dimensions? If original variables, I would repeat the need to clarify how the categorical variable was handled? I believe Gower dissimilarity matrix would consider the pairwise ecotype similarity so forest vs irrigated crops will have the same dissimilarity as irrigated vs non-irrigated crops. Is that a correction interpretation, and if so was that intended. Does that approach really help effectively use the ecotypes in this analysis, I believe not. More discussion of methods and their implication on results are needed.

We provide more details in the methods and summarize it here. Knowing that regional patterns in climate and land cover will be important for scaling $CH_4$ to the regional and national scale, after creating a dissimilarity matrix that was then scaled to two dimensions, we divided the US into clusters to evaluate representativeness using the first and second dimension from the MDS. This cluster analysis allowed us to summarize our results within a geographical context, an approach that has been used to delineate spatial sampling domains, to assess the spatial representativeness of networks, and to suggest arrangements of study sites (Sulkava et al. 2011; Kumar et al. 2016). We used a cluster analysis to determine the optimal number of clusters using the library *cluster* and the function pam in R (Reynolds et al. 2006; Schubert and Rousseeuw 2019; Schubert and Rousseeuw 2021). This approach uses the k-medoids algorithm, which partitions data set

into k groups or clusters and is a robust alternative to k-means clustering (Kaufman and Rousseeuw 2009). Each cluster is represented by one of the data points in the cluster named the cluster medoid. The medioid has the lowest average dissimilarity between it and all other objects in the cluster. The medoid can be considered a representative example of the members of that cluster. The k-medoid algorithm is less sensitive to noise and outliers, compared to k-means, because it uses medoids as cluster centers. The k-medoids algorithm requires the user to specify k, the number of clusters to be generated. A useful approach to determine the optimal number of clusters is the **silhouette** method. We fit an increasing number of clusters from 2 to 20 to construct a silhouette plot and choose the number of clusters that maximized the average silhouette width. Once we determined the number of clusters and the medoid of the cluster, we calculated the dissimilarity between every location within the cluster to the medoid to create a measure of how different each location was from the medoid condition of each cluster. We utilized the pointDistance function in the *raster* package, which provided a unit-less relative measure of dissimilarity that was determined by measuring the difference between the first and second dimensions produced by the isoMDS of each point in a cluster to the dimensions of the medioid.

- ○ Lines 222-224 states "We fit an increasing number of clusters from 2 to 20 to construct a silhouette plot and choose the number of clusters that maximizes the average silhouette width to determine an optimal number of clusters." However, the discussion section jumps to discuss k=10 clusters, but what about 2-9, 11-20? Why 10, why not some other number between 2-20?

We will added more detail to the results to show that the **silhouette** plots were used to determine the optimal number of clusters.

- ○ Lines 236-238 "To extrapolate the cluster and distance layers across the entire US beyond the 20,000-pixel subsample, we fit a Random Forest model with the package randomForest (Liaw and Wiener, 2002) to model the first and second MDS dimension using the ecotype and climate layers as predictors." I am struggling to understand what this statement means. A Random Model model is being fit to model first and second MDS dimension using ecotype and climate layers as predictors. BUT weren't MDS dimensions calculated using ecotype and climate layers in the first place. Is this Random Forest model a replacement for R MASS package. Why, this step makes no sense, unless I am missing something or this statement is somehow mis-written.

- ○ Lines 239-240 "We then created a Random Forest model of the cluster layer using the first and second dimension as the explanatory variables." Why

would you not just cluster the first and second MDs dimension, instead of creating a Random Forest model to identify the cluster layer. This seems like unnecessarily convoluted approach which really makes no sense and is addding methodological complexiy and model uncertainties.

To measure dissimilarity, we first randomly sampled  (n = 20,000 1-km pixels) the US, maintaining the distribution of ecotypes and climate to define dissimilarity between observations. Although there were more than 8 million sampling points available for the U.S, there are limits to the number of samples that can be analyzed by the functions used for the MDS analysis. This analysis was repeated 10 times to ensure that the 20,000 pixel subsample would produce similar results in the dimensions and clustering. For simplicity, we show the results of the first analysis.

To extrapolate the cluster and dissimilarity layers across the entire US beyond the 20,000-pixel subsample, we fit a Random Forest model with the package *randomForest* (Liaw and Wiener, 2002) to model the first and second dimensions using the ecotype and climate layers as predictors. We then created a Random Forest model of the cluster layer using the first and second dimension as the explanatory variables. All models were then projected spatially to produce a spatially explicit cluster layer and a dissimilarity layer beyond the 20,000 sample points that were used in the MDS analysis.  This allowed us to measure the importance of the original data on the first and second dimensions defined by the MDS and how the MDS leads to cluster and dissimilarity patterns. This step was essential to producing a spatially explicit cluster and dissimilarity layers for the entire US, since the MDS analysis limits the number of observations that can be analyzed.  We added more detail to the text and the results of the random forest model and the importance of the original data in explaining clustering and the first and second MDS dimensions.

Comments on results and discussion:

Results and discussion section is more about stating the results and is really lacking in discussion of results, why they were calculated and what they mean for the science question central to the study?

- Lines 260-263 "Latitude ($R^2 = 0.95$; $p < 0.001$), mean annual temperature ($R^2 = 0.84$; $p < 0.001$), maximum temperature ($R^2 = 0.83$; $p < 0.001$), vapor pressure deficit ($R^2 = 0.83$; $p < 0.001$), minimum temperature ($R^2 = 0.82$; $p < 0.001$), longitude ($R^2 = 0.63$; $p < 0.001$) had strong effects on clustering, whereas precipitation ($R^2 = 0.10$; $p < 0.001$), and ecotype ($R^2 = 0.03$; $p < 0.001$) showed low correlations." First, I am still not clear what these correlations are? Simply the fact that ecotype and precipitation has very low correlations for clustering is the huge red flag that these clusters are not appropriate for quantifying the representation of CH4 measurement sites. In the introduction section, authors have made strong arguments about importance of landcover, wetlands and agricultural use classifications. By their own measure, clusters that are insensitive to these landcover type are not appropriate for estimating CH4 fluxes. Purpose of study of

to identify cluster that represent CH4 and other GHG flues, and not to identify site that represent the clusters well. I believe the variable and clustering approach applied are not able to capture the heterogeneities on conditions that drive CO2 and CH4 fluxes, especially in wetlands, croplands and near aquatic sites.

These correlations were the average, not for clustering but for the first and second dimensions. We removed these results and replaced them with the variable importance from the random forest. It is important to note that the analysis done here  will not capture the heterogeneity of the conditions that drive $CH_4$ fluxes at the ecosystem scale. It is designed to evaluate the sampling intensity of research sites at the landscape scale. In the design of a network, this course resolution influences the capacity to scale ecosystem level results to the landscape, region, and to the national level, which is required for the development of budgets and emission strategies.

- Lines 270-272 "We found the size of the cluster is not correlated to the number of towers when all towers are included in the analysis but was slightly negatively correlated with the number of EC towers that include CH 4 measurements (Figure 2)." I am not sure such correlations are meaningful at all. What is the purpose of correlations (postive or negative)? **This was removed from the analysis. It was originally included to show the poor relationship between cluster size and the frequency of towers.  We agree with the review, this figure is unnecessary and is no longer relevant, since we are no longer including all towers.**
- Mediod locations of 10 clusters would be the theoretically optimal locations for locating an EC site. It would help to see a map of where these 10 locations and are perhaps a discussion of how well they appear to capture the local methane source/sinks on the ground. 10 is a small enough number to present a short and meaninful discussion to show the effectiveness of cluster medioid method.

[Figure]

**Figure 2. Cluster dissimilarity for the US. Inset: the distributions of dissimilarity across all clusters shown in a histogram, in which the line denotes the mean dissimilarity across all clusters.**

Figure 2 shows patterns in dissimilarity across the US with cyan locations having the lowest dissimilarity. We will add additional maps to the supplemental to highlight areas with the lowest dissimilarity. We also agree that it would be informative to show how towers in the medoid capture the local methane source/sinks on the ground. Unfortunately, towers are not currently distributed across cluster/medoid or IGBP to facilitate an evaluation of source sink potential. This highlights the limitations of existing measurement infrastructure which requires strategic augmentation to provide the most valuable information toward reducing uncertainties in future large-scale budget estimations. This analysis complements previous studies based on climatic or vegetation characteristics (Hargrove et al. 2003; Yang et al. 2008; Villarreal et al. 2018), and identifies regions within the US where gaps are limiting the development of upscaling techniques. To accurately understand the impact of climate and land cover change on biogenic $CH_4$ emissions, we need a long-term, calibrated, and strategic continental-scale $CH_4$ observatory network. Current gaps in existing measurement infrastructure limit our ability to capture the

spatial and temporal variations of biogenic $CH_4$ fluxes and therefore limit our ability to predict future $CH_4$ emissions. Maps of potential $CH_4$ emissions require land cover classification targeted at land cover types like wetlands that are important sources of $CH_4$ to the atmosphere. Aquatic ecosystems like streams and lakes as well as coastal ecosystems are significant and variable sources of $CH_4$ not well studied on a long-term basis. Through our analysis using climate, land cover, and location variables, we have identified priority areas to enhance research infrastructure to provide a more complete understanding of the $CH_4$ flux potential of ecosystem types in the US. For EC tower locations, dissimilarity coverage was lacking for clusters Na, W, and Nb, and currently clusters Na, W, Eb, and Nb are substantially under sampled. All aquatic sites are under sampled within each cluster. An enhanced network would allow for us to monitor both the response of $CH_4$ fluxes to climate and land use change as well as the impact of future policy interventions and mitigation strategies.

Few suggestions:

- Please consider including additional variables such as soil moisture, some measure of inundation, soil organic carbon to better capture the CH4 sources/sinks.

While measures of soil moisture, inundation, and soil organic carbon are important drivers of ecosystem $CH_4$ production and consumption, at the scale we are working on climatic conditions are more appropriate, as to not suggest we are able to represent those layers in a meaningful way at a 1 km resolution. The landscape representativeness analysis done here uses a coarser classification of ecosystem types that are more emblematic of regional disturbance regimes, resource availability, and factors that influence how ecosystems function, not the specific ecosystem type where the tower is situated. Chu et al., 2021 examined the land-cover composition and vegetation characteristics of 214 AmeriFlux tower site footprints. They found that most sites do not represent the dominant land-cover type of the ecosystems they exists within and when paired with common model-data integration approaches this mis-match introduces biases on the order of 4%–20% for EVI and 6%–20% for the dominant land cover percentage (Chu et al. 2021), making it essential to consider landscape characteristics in the design and evaluation of network infrastructure. Tower representativeness at the landscape scale is indicative of the capacity to upscale information by climate and the dominant ecosystems of locations within a landscape. We also calculate regional representativeness by the towers' vegetation type to understand the sampling intensity of each vegetation type within a cluster, which is also an essential component of scaling $CH_4$ fluxes (Knox et al., 2019).

Simplify the methodology and cluster the entire US and not a small 20,000 subsample to make the best use of information and variability captured in the data. Clustering + MDS + RF is unnecessarily complicated and perhaps hurt and not help the analysis.

To measure dissimilarity, we first randomly sampled (n = 20,000 1-km pixels) the US, maintaining the distribution of ecotypes and climate to define dissimilarity between observations. Although there were more than 8 million sampling points available for the U.S, there are limits to the number of samples that can be analyzed by the functions used for the MDS analysis. This

analysis was repeated 10 times to ensure that the 20,000 pixel subsample would produce similar results in the dimensions and clustering. For simplicity, we show the results of the first analysis.

To extrapolate the cluster and dissimilarity layers across the entire US beyond the 20,000-pixel subsample, we fit a Random Forest model with the package *randomForest* (Liaw and Wiener, 2002) to model the first and second dimensions using the ecotype and climate layers as predictors. We then created a Random Forest model of the cluster layer using the first and second dimension as the explanatory variables. All models were then projected spatially to produce a spatially explicit cluster layer and a dissimilarity layer beyond the 20,000 sample points that were used in the MDS analysis.  This allowed us to measure the importance of the original data on the first and second dimensions defined by the MDS and how the MDS leads to cluster and dissimilarity patterns. This step was essential to producing a spatially explicit cluster and dissimilarity layers for the entire US, since the MDS analysis limits the number of observations that can be analyzed.  We added more detail to the text and the results of the random forest model and the importance of the original data in explaining clustering and the first and second MDS dimensions.

- It would be of more value to consider the operational vs non-operational status of the EC sites in the analysis, so the results can inform actionable decisions.

Thank you for this comment. We made changes to the towers used in this analysis. Although we have information on the location of existing EC tower infrastructure that is a part of AmeriFlux (N=223), NEON (N=47), and known, independent PI-managed sites (n=141). We focus this analysis on the towers measuring $CH_4$ (n=100) and we distinguish between towers providing data to Ameriflux (yes =49, no = 51) and tower activity (active = 70; inactive = 30). We understand that additional towers exist within the U.S, but because these towers are not reporting or providing data to the flux community, we cannot include them in this analysis.

**Table 4. The R $_{cluster}$ for $CH_4$ towers that are active and providing data to Ameriflux, the R $_{cluster}$ for all active $CH_4$ towers and the R $_{cluster}$ for all active towers in addition to NEON towers.**

| Cluster | $CH_4$ Towers (Data Providing) | $CH_4$ Towers (All) | NEON Towers |
|---|---|---|---|
| Na | 3.0 | 34.9 | 35.5 |
| NW | - | 0.1 | 26.3 |
| NEb | 19.8 | 60.6 | 65.9 |
| Ea | 0.01 | 63.1 | 89.4 |
| Eb | 88.1 | 88.1 | 88.1 |
| SW | 2.0 | 3.3 | 17.3 |
| W | 0.01 | 0.01 | 38.8 |
| NEa | 79.3 | 79.3 | 79.3 |
| Nb | 21.3 | 21.3 | 21.3 |
| SE | - | 23.6 | 50.8 |

There were important gains in TR $_{cluster}$ when considering all $CH_4$ towers regardless of if they were providing data to Ameriflux (Table 4 and Figure 4). The clusters with substantial gains in representativeness (> 10%) include Na, NEb, Ea, and the SE.  The TR $_{cluster}$ of the NW, Ea, SW, W, and the SE would be enhanced by more than 10% with the addition of $CH_4$ instrumentation at NEON sites.

---

## Author Response (AR2)

Thank you for your thoughtful comments and concerns about the methodology, in particular use of MDS + cluster analysis +Random Forest. In response to your concerns we added details/limitations of the cluster approach and an Appendix on the cluster solutions.

[revised manuscript text omitted]

**Supplement:**

We evaluated the stability of the cluster solution presented (Figure 2) by comparing it to the cluster solutions of 6 example cases (Table S1). The example cases (Samples 1-5) use the same analysis approach as the presented solution (Figure 2). While Sample 1 is from a systematically random sample, samples 2-5 are from a randomly sampled dataset. To understand the impacts of subsampling, we derived a cluster solution for each sample and compared it to the standard solution (Figure 2). The package *bigmds* uses the divide-and-conquer MDS approach (Delicado and Pachon-Garcia 2020). This algorithm partitions the data into subsamples (n=60,000), where classical methods can work. In order to align all the solutions, the Procrustes formula is used (Borg and Groenen 2005). Following the MDS we used the kmeans function to cluster the bigmds (Forgy 1965; Hartigan and Wong 1979; Lloyd 1982).

**Table S1: Example cases to test the stability of the cluster solution.**

| Example | Description | Sample Size | R Packages |
|---------|-------------|-------------|------------|
| Sample 1 | Systematic random sample across all climate and land cover classes. | 20,000 | *kmed. MASS, PAM* |
| Sample 2 | Random Sample | 20,000 | *kmed. MASS, PAM* |
| Sample 2 | Random Sample | 20,000 | *kmed. MASS, PAM* |
| Sample 3 | Random Sample | 20,000 | *kmed. MASS, PAM* |
| Sample 4 | Random Sample | 20,000 | *kmed. MASS, PAM* |
| Sample 5 | Random Sample | 20,000 | *kmed. MASS, PAM* |

| BigMDS | divide_conquer_mds | 8,268,498 | *bigmds, stats* |

To compare cluster solutions we measured the stability of a cluster (Figure S3 and Figure S4a.). Stability was measured as the consistency of clustering within the presented solution/standard (Figure 2). This approach is not influenced by the cluster label changing, but by the presence of multiple clusters occurring within the standard cluster. A stability of 100% indicates that all samples within the standard cluster belong to the same cluster in the sample solution.

Regardless of the sampling, similar clustering patterns were obtained (Figure S4) and the mean stability was 82% across the six examples explored (Appendix. 2). The mean stability of the bigmds example was 65%. The cluster stability was low between neighboring clusters NEb and NEa, and between Eb and the SE cluster. Altogether these results show that failing to subsample the data properly leads to declines in cluster stability and changes in the clustering method can lead to differences in the final cluster solution (Figure S4).

[Figure]

**Figure S3. The cluster solution for a. sample 1, b. sample 2, c. sample 3, d. sample 4, e. sample 5, and d. the Bigmds (Table S1).**

[Figure]

**Figure S4. The stability of the example cases compared to the standard cluster solution (Figure 2). The mean stability is 82% across all cluster solutions. The gray region was not sampled by sample 3.**